# Integrated omics in *Drosophila* uncover a circadian kinome

Chenwei Wang[1,4], Ke Shui [1,4], Shanshan Ma[1], Shaofeng Lin[1], Ying Zhang [1], Bo Wen [2], Wankun Deng[1], Haodong Xu[1], Hui Hu[1], Anyuan Guo [1], Yu Xue [1,3✉] & Luoying Zhang[1,3✉]

Most organisms on the earth exhibit circadian rhythms in behavior and physiology, which are driven by endogenous clocks. Phosphorylation plays a central role in timing the clock, but how this contributes to overt rhythms is unclear. Here we conduct phosphoproteomics in conjunction with transcriptomic and proteomic profiling using fly heads. By developing a pipeline for integrating multi-omics data, we identify 789 (~17%) phosphorylation sites with circadian oscillations. We predict 27 potential circadian kinases to participate in phosphorylating these sites, including 7 previously known to function in the clock. We screen the remaining 20 kinases for effects on circadian rhythms and find an additional 3 to be involved in regulating locomotor rhythm. We re-construct a signal web that includes the 10 circadian kinases and identify GASKET as a potentially important regulator. Taken together, we uncover a circadian kinome that potentially shapes the temporal pattern of the entire circadian molecular landscapes.

[1] Key Laboratory of Molecular Biophysics of Ministry of Education, Hubei Bioinformatics and Molecular Imaging Key Laboratory, Center for Artificial Intelligence Biology, College of Life Science and Technology, Huazhong University of Science and Technology, Wuhan, Hubei 430074, China. [2] Department of Molecular and Human Genetics, Lester and Sue Smith Breast Center, Baylor College of Medicine, Houston, TX 77030, USA. [3] Institute of Brain Research, Huazhong University of Science and Technology, Wuhan, Hubei 430074, China. [4] These authors contributed equally: Chenwei Wang, Ke Shui. ✉email: xueyu@hust.edu.cn; zhangluoying@hust.edu.cn

Circadian clocks drive daily, or circadian rhythms in a myriad of biological processes. Disruptions of these rhythms are associated with various diseases and disorders, such as cancers, metabolic disorders, and mood disorders[1]. Underlying the overt rhythms are molecular oscillations at multiple levels, including cyclic regulations of the transcriptome, proteome, and post-translational modifications (PTMs)[2–5]. However, it remains largely unclear how circadian clocks drive these integrated rhythms.

The molecular clock consists of a series of transcriptional and translational feedback loops that are relatively conserved[2]. In *Drosophila*, two transcription factors, CLOCK (CLK) and CYCLE (CYC) are at the center of the loops. CLK and CYC dimerize and activate the transcription of *period* (*per*) and *timeless* (*tim*) via E-box elements in the genome[6]. PER and TIM proteins accumulate in the cytoplasm, form a complex, and enter the nucleus, where they suppress the transcriptional activities of CLK/CYC and thus their own transcription. During this process, PTMs influence the stability and nuclear translocation of PER and TIM, leading ultimately to their degradation that enables CLK/CYC to start a new round of transcription, thus completing a cycle[2]. There are also a few accessory loops that result in rhythmic transcription of *clk*. The time it takes for these feedback loops to operate once is ~24 h, and PTMs, especially phosphorylation, play a central role in timing the molecular clock[6–8]. Moreover, rhythmic phosphorylation has been suggested to be a fundamental part of the clock and at the heart of molecular oscillations, as it has been demonstrated in cyanobacteria and *Neurospora* that a purely phosphorylation-based clock is sufficient to drive circadian cycling[9,10].

With the advances in high-throughput mass spectrometry, time series analysis of proteomics and phosphoproteomics have been conducted in mouse livers, demonstrating that rhythmic phosphorylation is not limited to the core clock[3–5]. About 25% of all phosphorylation sites (p-sites) in mouse liver exhibit robust circadian oscillations[3]. How these oscillations in phosphorylation are regulated is unknown.

Here, we conduct a multi-omics profiling to measure circadian oscillations in transcriptomes, proteomes and phosphoproteomes in fly heads. We develop an efficient pipeline for computationally integrating circadian multi-omics data (iCMod) to acquire normalized circadian p-sites (NCPs) that are oscillating in a circadian manner truly owing to rhythmic phosphorylation/dephosphorylation events. In total, we quantify 4686 p-sites with high confidence from wild-type (WT) fly heads, among which 789 (~17%) NCPs characterized from 431 proteins display circadian oscillation. Most of these rhythms are dampened in mutants lacking core clock gene *per* (*per[0]*), implicating these rhythms are driven by the molecular clock. We predict that 27 protein kinases might be involved in regulating circadian rhythms by preferentially phosphorylating these NCPs, including seven kinases already known to play essential roles in the core clock. To validate our predictions, we test the remaining 20 kinases and discover *gasket* (*gskt*), *Downstream of raf1* (*Dsor1*) and *casein kinase I alpha* (*CKIalpha*) to be participating in determining the period and/or power of locomotor rhythm. Computational analysis reveals that these 10 kinases involved in locomotor rhythm may contribute to global oscillations not only at phosphorylation level, but also at mRNA and protein levels with GSKT as a potentially critical regulator of the signaling cascades. We further examine the function of GSKT within the clockwork and find it acts to reduce TIM protein but not mRNA level. Taken together, our results unveil a kinome that is potentially involved in shaping the entire circadian molecular landscapes, as well as intricate interactions among the kinases and their substrates that ultimately impinge on locomotor rhythm.

## Results

**Quantifying multi-omics data under circadian cycles.** To investigate global phosphorylation in flies, we conducted quantitative proteomics and phosphoproteomics using the Tandem Mass Tag (TMT) labeling and liquid chromatography-tandem mass spectrometry (LC-MS/MS) on WT and *per[0]* fly heads collected at 3 h intervals on 2 days under constant darkness (DD) condition (Fig. 1a). Altogether we identified 61,460 non-phosphorylated peptides and 12,465 phosphopeptides from 32 samples. The majority of the peptides (35,280; 57.40%) and phosphopeptides (8193; 65.73%) could be matched with ≥2 spectral counts, whereas the average spectral counts were 2.5 and 4.4 for all peptides and phosphopeptides, respectively (Fig. 1b). We next mapped non-phosphorylated peptides to their corresponding protein sequences, and obtained 5998 and 6034 proteins in WT and *per[0]* flies, respectively (Supplementary Data 1). Only 14.87% (912) of 6134 quantified proteins were assigned with one matched peptide, with an average number of 8.6 quantified peptides per protein (Fig. 1c). We also mapped phosphopeptides to full-length protein sequences and in total obtained 3295 phosphoproteins with 14,946 non-redundant p-sites from all 32 samples with an average p-site localization probability of 0.91, including 12,399 p-Ser (82.96%), 2458 p-Thr (16.45%), and 89 p-Tyr (0.60%) sites (Fig. 1d, e and Supplementary Data 1). We compared the p-sites identified here with eight public databases, including dbPAF[11], dbPTM[12], Phospho.ELM[13], PHOSIDA[14], PhosphoPep[15], PhosphoSitePlus[16], SysPTM[17], and UniProt[18]. Only 37.56% p-sites quantified in this study were annotated and included in at least one phosphorylation database, whereas up to 9333 p-sites have never been reported (Fig. 1f). By using two-sided hypergeometric test, the enrichment analysis of Gene Ontology (GO) terms revealed that proteins expressed in the head were mainly involved in neurotransmitter secretion, translation, transport, and splicing, whereas phosphorylation is enriched in pathways that regulate GTPase activity, olfactory learning, chemical synaptic transmission, and intracellular signaling, as well as protein phosphorylation (Supplementary Fig. 1a).

To further validate the proteomic and phosphoproteomic data sets, we conducted transcriptome profiling of WT and *per[0]* fly heads collected in DD by RNA sequencing (RNA-seq) (Fig. 1a). Over $9.4 \times 10^8$ reads were sequenced in all 32 samples, with an average of $3.1 \times 10^7$ and $2.9 \times 10^7$ reads in WT and *per[0]*, respectively (Supplementary Fig. 1b). After reads mapping and transcript assembly, there are 15,280 and 14,760 mappable protein-coding genes identified in WT and *per[0]* flies, respectively, which occupy 70.84% of the fly protein-coding transcriptome (Supplementary Fig. 1c and Supplementary Data 1). Fragments Per Kilobase of exon per Million fragments mapped (FPKM) values were calculated for the quantification of individual mRNAs, and the average FPKM values are 114 and 118 for WT and *per[0]* flies, respectively (Supplementary Fig. 1d and Supplementary Data 2). Similar to proteomic data, mRNAs expressed in the head are also enriched in the splicing process, as well as axon guidance, development and transcription (Supplementary Fig. 1a).

To ensure data quality, only proteins and p-sites quantified in all 16 samples of WT or *per[0]* flies were retained (Supplementary Fig. 1e–g). In total, there are 4537 proteins and 5724 p-sites quantified in all WT flies, whereas 4561 proteins and 5739 p-sites were quantified in all *per[0]* samples. The multi-omics measurements show high reproducibility, with Spearman's rank correlation coefficients of 0.99, 0.95, and 0.85 at mRNA expression, protein expression, and phosphorylation levels for the two cycles monitored, respectively (Fig. 1g). After normalization of the proteomic and phosphoproteomic data (Supplementary Fig. 1h), we analyzed the correlation among the multi-omics data, and found the correlation of temporal variation between proteome

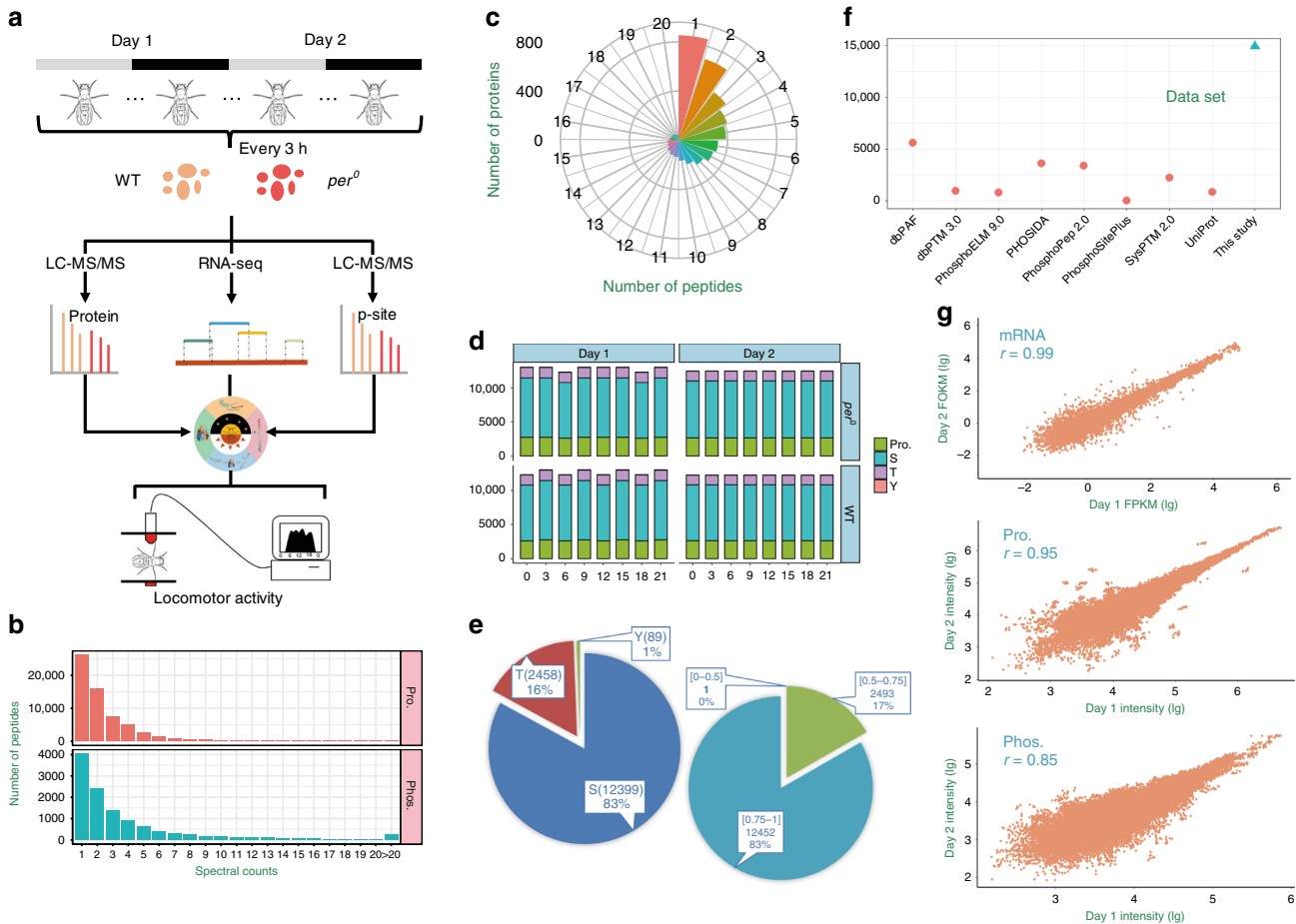

**Fig. 1 Circadian multi-omics profiling of fly heads. a** WT and *per⁰* fly heads were collected at 3 h intervals on 2 days in DD. LC-MS/MS-based proteomics and phosphoproteomics as well as RNA-seq-based transcriptomics were conducted. An integrative pipeline iCMod was implemented for analyzing the multi-omics data. NCPs were detected and their corresponding protein kinases were predicted. Locomotor rhythm analysis was employed for validation of the predicted kinases. **b** The distributions of raw MS/MS spectral counts of peptides and phosphopeptides quantified from proteomics and phosphoproteomics data, respectively. **c** The distribution of peptide numbers quantified from proteomics data. **d** The number of phosphoproteins, as well as p-Ser, p-Thr, and p-Tyr residues identified in each sample. **e** The distribution of the amino-acid residues (left) and the assigned localization probability (right) for all detected p-sites. **f** Comparison of p-sites detected in this study with known p-sites curated in public databases. **g** The Spearman's rank correlations of transcriptomes, proteomes, and phosphoproteomes detected on Day 1 and Day 2, respectively.

and phosphoproteome is much higher than that between transcriptome and proteome (Supplementary Figs. 2, 3, 4a and Supplementary Note 1). These results suggest that phosphorylation has a major role in regulating the temporal alterations of protein level.

**Employing iCMod for integrating circadian multi-omics data.** In this work, we implemented a pipeline termed iCMod to integrate circadian multi-omics data, and to accurately predict mRNAs, proteins and p-sites with circadian oscillation (Fig. 2). For a protein detected by proteomics/phosphoproteomics analysis, we believe its corresponding mRNA should be readily detectable in the transcriptomic data. For 16.70% and 16.80% of quantified proteins from WT and *per⁰* flies, respectively, we observed that their corresponding mRNAs are expressed at a low level with FPKM < 1 (Supplementary Fig. 4b). Previous study has demonstrated that mRNAs with FPKM < 1 cannot be reliably determined to be expressed[19], and excluding weakly expressed transcripts enhances the reliability of protein identification[20]. Therefore, we constructed customized protein sequence databases using only mRNAs with relatively high expression level (FPKM ≥ 1 in at least one sample per batch). Peptides and phosphopeptides

were then identified from proteomic and phosphoproteomic spectra by searching the reference databases, respectively (Fig. 2 and Supplementary Data 3–5).

We employed ARSER[21] to identify genes with circadian expression from mRNAs with FPKM ≥ 1 detected in at least one of the 16 samples of WT and *per⁰* flies, respectively. For proteomic data, the average intensity value of all proteins was normalized to 1 for each sample. ARSER was subsequently used to identify proteins that oscillate in abundance. As the phosphoproteome shows strong positive correlation of temporal expression with the proteome (Supplementary Fig. 2a), it is possible that for many p-sites, oscillation of phosphorylation level is a result of oscillation in protein level. Therefore, we normalized the phosphoproteomic data by calculating the ratio of raw phosphorylation abundance vs. raw protein abundance for each p-site to acquire normalized phosphorylation level. We then used ARSER to identify NCPs, which reflect true oscillations at the phosphorylation level (Fig. 2). We adopted a previously developed tool, Group-based Prediction System 2.1 (GPS 2.1), to predict site-specific kinase–substrate relations (ssKSRs) for the p-sites[22]. This is followed by a two-sided hypergeometric test to determine protein kinases with substrates enriched for NCPs, which we referred to as potential circadian kinases.

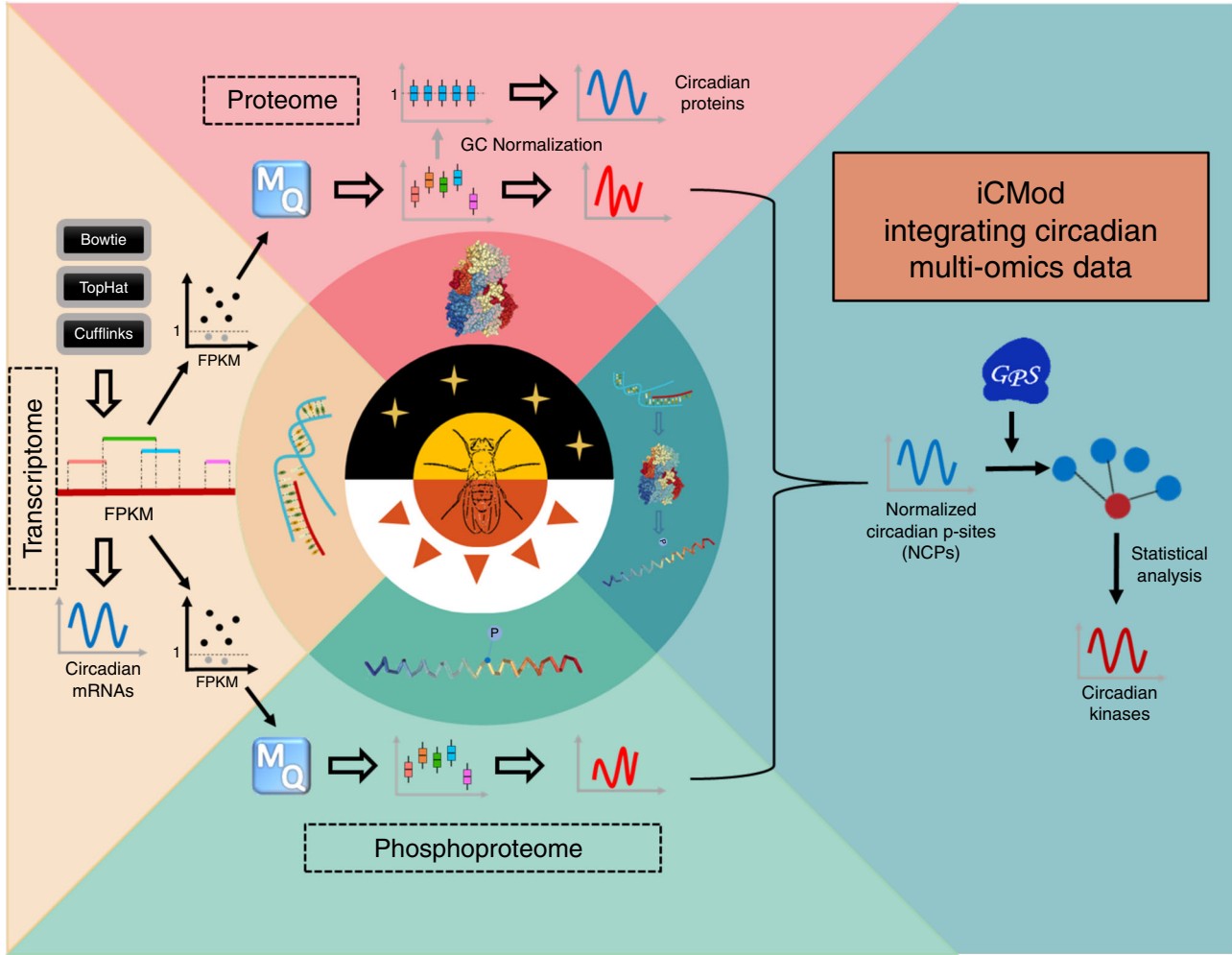

**Fig. 2 The iCMod pipeline for analysis of multi-omics data and prediction of circadian protein kinases.** We constructed a reference database for MS/MS searching containing only protein sequences with corresponding mRNA levels of FPKM ≥ 1 in at least one sample. After the database search for raw proteomic and phosphoproteomic spectra, the intensity values of proteins were normalized to one for each sample. Phosphorylation level was normalized to the corresponding protein level. ARSER was used to identify cycling mRNAs, proteins, and NCPs. GPS 2.1 was subsequently employed to predict ssKSRs, followed by a two-sided hypergeometric test to predict potential circadian kinases.

**Molecular oscillation landscapes regulated by PER**. After processing the omics data sets with iCMod, we identified 661 (6.67%) mRNAs, 620 (16.42%) proteins, and 789 (16.84%) p-sites to be significantly oscillating in WT fly heads (Fig. 3a and Supplementary Data 6). In contrast with the commonly used approach, which does not remove mRNAs with FPKM < 1 or normalize phosphorylation abundance to protein abundance, iCMod enabled more accurate identification of cycling molecules (Fig. 3a, Supplementary Figs. 5, 6, and Supplementary Note 2). We compared our circadian multi-omics data with circadian gene database (CGDB, http://cgdb.biocuckoo.org/)[23] which includes 2768 *Drosophila melanogaster* genes with mRNAs reported to be cycling and found these genes to be enriched only in our cyclic mRNA data set but not in the cyclic protein or phosphoprotein data sets (Fig. 3b). Moreover, we compared our circadian omics data with an oscillatory translatome data set acquired by ribosomal profiling of fly heads[24], and found that genes rhythmically translated are statistically enriched both in our cyclic mRNA and protein data sets, but not in cyclic phosphoprotein data set (Fig. 3b). Taken together, iCMod integrates multiple types of omics data, and determines circadian oscillations at multiple levels with higher confidence and accuracy.

To test whether these molecular oscillations are driven by the core molecular clock, we used iCMod to process multi-omics data from *per⁰* heads (Fig. 3c). We observed that *per⁰* mutation abolishes the cycling of 93.95% of mRNAs, 84.52% of proteins and 87.96% of p-sites, indicating that the majority of the molecular oscillations are controlled by PER, which means they are likely driven by the molecular clock. Moreover, we calculated the peak times for the molecular oscillations and found that cycling transcripts tend to peak at CT12 (Fig. 3d). The peak times for cycling proteins appear to be rather evenly distributed throughout the day, whereas NCPs tend to peak more during the subjective night from CT12 to CT0 (Fig. 3d, e). The amplitude of these oscillations is relatively low, with the amplitude of cycling at phosphorylation level slightly higher than that at protein level (Fig. 3f). To understand the potential function of these molecular oscillations, GO-based enrichment analysis was performed (Fig. 3g). Cycling mRNAs are enriched in light and visual signaling pathways, while cycling proteins are enriched in processes involved in Wnt regulation and endocytosis/exocytosis. NCPs are enriched in pathways that regulate sarcomere organization and neural development. We did not identify any published cycling p-sites among the NCPs, which is probably because the published sites are located on core clock

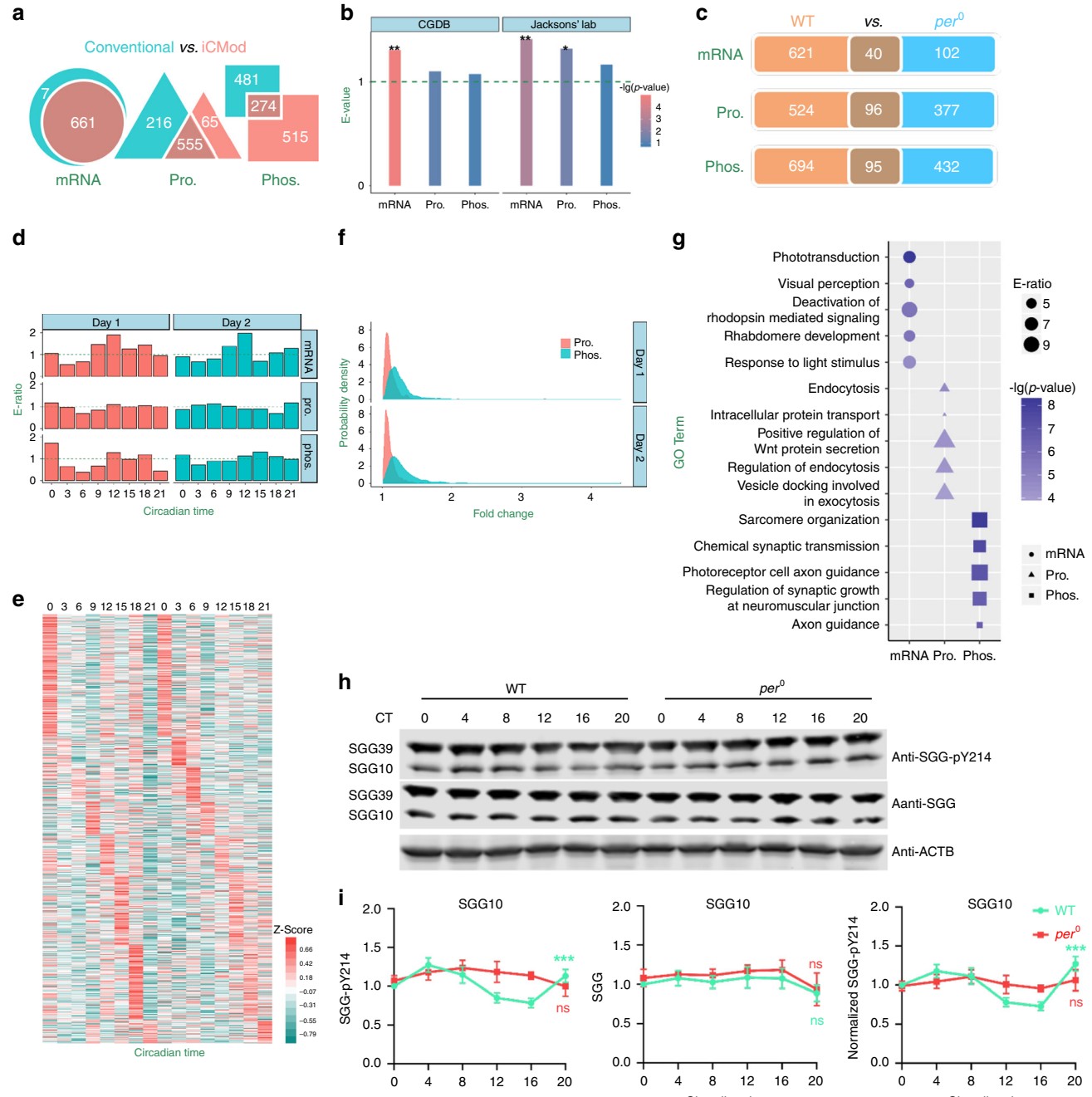

**Fig. 3 Molecular oscillation landscapes uncovered by iCMod. a** Cycling mRNAs, proteins, and p-sites identified by iCMod vs. commonly used approach. **b** The enrichment analyses of 2768 transcriptionally rhythmic genes collected in CGDB[23] and 1255 translationally oscillated mRNAs obtained from a previously reported ribosomal profiling against circadian mRNAs, proteins, and phosphoproteins detected by iCMod (two-sided hypergeometric test, from left to right, p value = 0.00001, 0.0614, 0.1299, 0.0013, 0.0305, and 0.1572; **p value < 0.01, *p value < 0.05). **c** Cycling mRNAs, proteins and NCPs identified in WT and per[0] flies. **d** The enrichment ratio of NCPs that peak at a certain time point. **e** A hierarchical clustering of the NCPs ordered by the phase of the oscillation. Values of each NCP in all analyzed samples (column) are color coded based on the intensities. Colors indicate low (green) and high (red) Z scores. **f** The distribution of fold-changes for cycling proteins and NCPs. **g** The GO-based enrichment analysis of cycling mRNAs, proteins and NCPs (Two-sided hypergeometric test, p value of mRNA < 2.2 × 10$^{-5}$, p value of Pro. < 1.4 × 10$^{-4}$ and p value of Phos. < 3.1 × 10$^{-7}$). **h** Western blots of proteins from whole-head extracts of WT and per[0] collected at the indicated circadian time. ACTB was used as a loading control. SGG10 and SGG39 indicate different isoforms of SGG. Repeated five to seven times independently with similar results. **i** Quantification of SGG pY214 (n = 7), SGG protein (n = 5) and normalized SGG pY214 (n = 7) levels of blots in **h**. Error bars represent standard error of the mean (SEM). SGG pY214 of WT, one-way ANOVA, ***p value = 0.00033; SGG pY214, one-way ANOVA of WT, ***p value = 0.00002; ns, not significant. Source data are provided as a Source Data file.

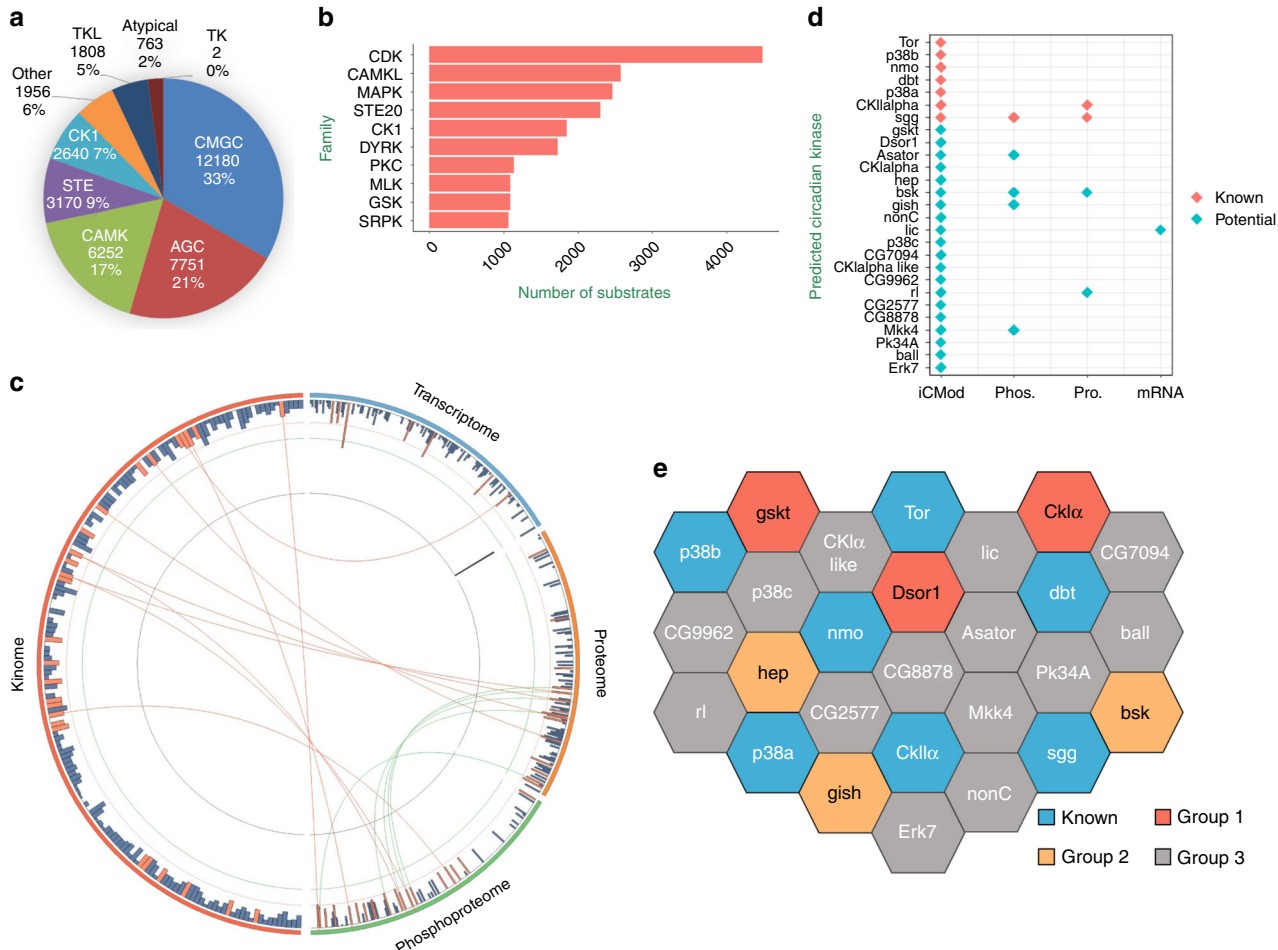

**Fig. 4 Computational prediction of a circadian kinome. a** The distribution of protein kinase groups predicted to phosphorylate NCPs. The top five groups of kinases are CMGC (cyclin-dependent kinase [CDK], mitogen-activated protein kinase [MAPK], glycogen synthase kinase [GSK3], CDC-like kinase [CLK]), AGC (protein kinase A, G, and C), CAMK (calcium/calmodulin-dependent protein kinase), STE (Ste20 kinase), and CK1 (casein kinase I). **b** The top 10 kinase families predicted to phosphorylate NCPs. **c** Each one of the four parts of the circle includes all 169 kinases that can be predicted by GPS 2.1. Each bar indicates a kinase. Kinases that oscillate at mRNA, protein or phosphorylation level are indicated in orange. Predicted circadian kinases are also indicated in orange. Among the kinases labeled in orange, the same kinase is connected by a line. **d** 27 potential circadian kinases predicted by iCMod, including seven that are already known to regulate the clock (red) and 20 that potentially participate in circadian regulation (green). Predicted circadian kinases that oscillate at mRNA, protein, or phosphorylation level are indicated. **e** Seven kinases previously known to regulate the clock are labeled in blue. Three kinases are found to be involved in modulating locomotor rhythm (labeled in red), and three more are found to be potentially involved (labeled in orange). The remaining 14 predicted circadian kinases are labeled in gray.

proteins such as PER, TIM and CLK[6]. These core clock proteins are expressed at low levels in whole-head extracts and thus we were not able to detect the reported phosphorylation rhythms on these proteins. We tried to validate some of our NCPs with antibodies available and was able to observe significant temporal variation at tyrosine 214 of SHAGGY10 (SGG10) isoform of the SGG protein which is eliminated by *per*[0] mutation (Fig. 3h, i and Supplementary Fig. 7)[25]. Taken together, our analyses identified substantial molecular oscillations occurring at the mRNA, protein, and phosphorylation levels that are largely controlled by PER and enriched in distinct biological pathways, which implies different biological processes favor different levels of circadian regulation.

**Prediction and validation of a circadian kinome**. We used GPS to analyze the 789 NCPs identified in WT flies, and acquired 36,522 potential ssKSRs between 153 protein kinases and 778 phosphorylated substrates. We found that 98.10% of all identified NCPs were predicted with ≥2 kinases, indicating a complex kinase–substrate phosphorylation network involved in circadian

regulation. The top five groups of kinases are responsible for the modification of ~87% of total phosphorylation events, whereas the top 10 kinase families identified carry out ~54% of total phosphorylation events (Fig. 4a, b). Next, we performed a two-sided hypergeometric test to analyze the enrichment of NCPs among all predicted substrates for each kinase. Altogether, we predicted 27 potential circadian kinases that display significant enrichment in NCPs, which include seven kinases already known to be involved in circadian regulation in flies (Fig. 4c, d)[6,26,27]. Eight of these kinases exhibit cycling at mRNA, protein and/or phosphorylation level, implicating a potential role as an instructive signal in the circadian system.

Among the predicted circadian kinases, there are 20 that have not been reported to participate in *Drosophila* circadian regulation (Fig. 4d). Therefore, we validated these kinases by testing whether they play a role in modulating fly locomotor rhythms. We found six kinases to be involved or potentially involved in rhythm regulation (i.e., when genetically manipulated, the target kinase can alter the period by at least 1 h or reduce the power of rhythm

by 50% or more), including *gskt, Dsor1, bsk, gish, hep,* and *CKIalpha* (Fig. 4e). *gskt* is generated by retroposition of *sgg*, the fly homolog of mammalian glycogen synthase kinase-3β (GSK3β)[28]. *Dsor1* encodes the *Drosophila* MEK, which is activated by receptor tyrosine kinases and phosphorylates extracellular-signal-regulated kinase[29]. *bsk* encodes the fly homolog of mammalian c-Jun amino terminal kinase (DJNK) and is known to be phosphorylated and activated by JNK kinase HEP[30]. *gish* encodes CKIγ and together with CKIalpha belong to casein kinase I (CKI) family[31].

We first examined the effects of these kinases on circadian regulation by knocking down each kinase in all clock cells using a *tim*GAL4 driver or in circadian neurons along with a few other brain regions with a *cryptochrome (cry)*GAL4-16[32]. We tested 89 RNA interference (RNAi) lines, which produce double-stranded RNA hairpin structures that trigger sequence-specific post-transcriptional silencing and RNAi responses[33,34]. We found that knocking down *gskt, Dsor1,* and CG7094 with at least two independent RNAi lines lead to 1–3 h lengthening of circadian period, whereas knocking down *Dsor1* also substantially reduces power and rhythmicity (Fig. 5a, b and Supplementary Data 7). Knocking down *CKIalpha* with one RNAi line almost completely abolishes the rhythm, whereas with another RNAi line leads to 2–3 h longer period. We then verified that the mRNA levels of the target genes were indeed reduced by RNAi (Supplementary Fig. 8a). Although two independent CG7094 RNAi lines exhibit lengthened period, only one of them shows significant mRNA

reduction (Fig. 5a, b and Supplementary Fig. 8a). We reasoned that the period lengthening effect of the other RNAi line may be owing to off-target effects, and thus did not consider CG7094 as a positive hit. There are several genes with only one RNAi line that exhibits substantial changes in period or power of the rhythm, which may also be owing to off-target effects of this particular RNAi, so we did not consider these genes as positive hits either (Supplementary Data 7). We tested nine overexpression lines and found that expressing a WT or dominant negative form of BSK in circadian neurons results in 1–3 h longer period, whereas overexpressing two constitutively active forms of HEP almost completely abolishes the rhythm (Fig. 5c)[35]. Overexpressing a WT GISH shortens the period by ~1.5 h whereas expressing a kinase dead form of GISH results in ~1.5 h lengthening of the period[36]. Overexpressing a WT CK1alpha substantially reduces the power of the rhythm. At last, we tested 33 mutant or potential mutant lines and found three mutants displayed reduced power or lengthened period (Supplementary Figs. 8b, 9, and Supplementary Data 7). However, after backcrossing these lines onto an isogenic background and/or testing additional alleles, we observed either no phenotype or weaker phenotype, which means the phenotypes previously observed are likely due to genetic background differences rather than defects caused by the mutations (Supplementary Fig. 8c and Supplementary Data 7).

Based on the strength of circadian phenotypes observed, we classified the remaining 20 predicted circadian kinases into three

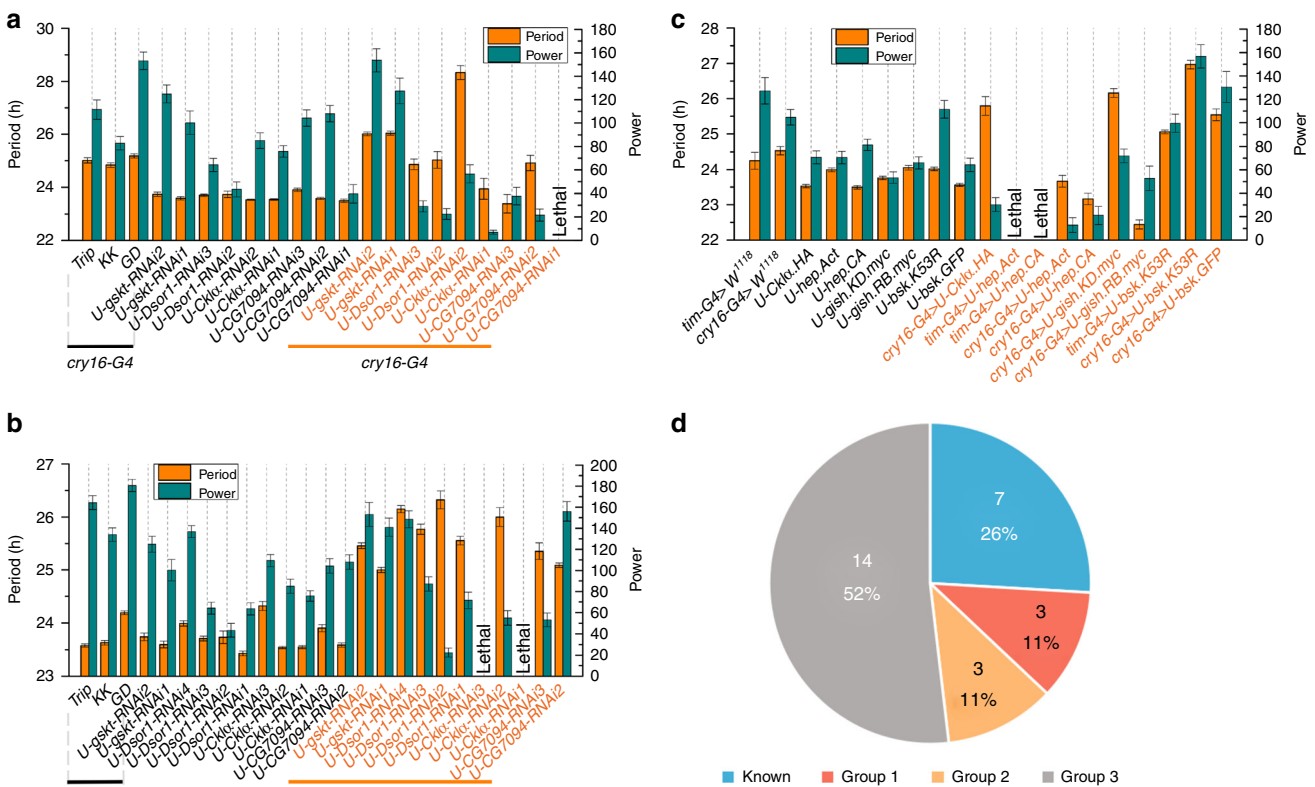

**Fig. 5 Identification of kinases that regulate locomotor rhythm. a** Period and power of DD locomotor rhythms of flies with *gskt, Dsor1, CK1α,* and CG7094 knocked down using *cry-GAL4-16,* along with the controls. *dicer2 (dcr2)* is co-expressed to enhance the effects of RNAi (From left to right, *n* = 43, 74, 50, 27, 26, 48, 42, 61, 72, 43, 45, 20, 37, 36, 36, 43, 45, 53, 22, and 42). **b** Period and power of DD locomotor rhythms of flies with *gskt, Dsor1, CK1α,* and CG7094 knocked down using *tim*GAL4, along with the controls (From left to right, *n* = 52, 61, 74, 27, 26, 47, 48, 42, 47, 59, 61, 72, 43, 45, 37, 26, 44, 45, 30, 30, 25, 38, and 45). **c** Period and power of DD locomotor rhythms of flies overexpressing WT or mutant forms of *hep, bsk, CK1α,* and *gish,* along with the controls (From left to right, *n* = 28, 44, 30, 60, 29, 43, 41, 30, and 31). All of the experimental groups show significant difference compared with the corresponding controls. Error bars represent SEM. One-way ANOVA, individual *p* values are shown in Supplementary Data 7. G4, GAL4; U, UAS. **d** Percentage of predicted circadian kinases that are previously known to regulate the clock (blue), identified here to be involved (red) or potentially involved (orange) in circadian regulation, or not yet validated (gray).

groups. Group 1 includes GSKT, DSOR1, and CKIalpha, the phenotypes of which were confirmed by multiple independent RNAi lines and thus are highly likely to regulate the clock. Group 2 includes GISH, BSK, and HEP, which only demonstrated phenotypes when overexpressed and thus are potential regulators of the clock. The rest of the predicted circadian kinases belong to Group 3, as they showed no clear evidence of involvement in locomotor rhythm modulation. All in all, nearly half (48%) of the predicted circadian kinases appear to play (or potentially play) a role in regulating locomotor rhythms, demonstrating the power of iCMod in identifying kinases that function in the circadian system and revealing insights regarding circadian control of phosphorylation (Fig. 5d).

**A signal web that controls molecular and locomotor rhythm.** According to our predicted ssKSRs and published literature, the seven known circadian kinases and 3 Group 1 kinases that regulate locomotor rhythm can form a giant web that integrates our circadian omics data from multiple levels. First of all, the activities of these 10 kinases can account for 81.6% of the NCPs identified (Supplementary Fig. 10 and Supplementary Data 8). For proteins that show circadian cycling in abundance, 76 contain NCPs predicted to be phosphorylated by one of the 10 kinases, whereas 429 interact with at least one other protein that contains NCPs phosphorylated by one of the 10 kinases. Therefore, the activities of these kinases could potentially explain 81.5% of the circadian proteome. Moreover, computational analysis predicted that the transcription of all genes oscillating at mRNA level is potentially activated by 33 transcription factors, 30 of which are predicted to be phosphorylated by at least one of these 10 kinases. These

transcription factors include core clock proteins CLK, CYC, VRILLE (VRI), and Par Domain Protein 1 (PDP1) as well as KAYAK (KAY) and MEF2, which have been shown to regulate the clock[37,38]. Notably, we also observe SERPENT (SRP) among the transcription factors, which is believed to act in synergy with CLK/CYC to orchestrate tissue-specific outputs[39]. Taken together, these results suggest that these kinases not only control rhythms at phosphorylation level, but also contribute to oscillation at mRNA level by modifying transcription factors and oscillation at protein level by directly phosphorylating the target proteins or indirect modulation via protein–protein interactions (PPIs).

Based on our predictions and published data, we further extracted a network of signaling cascades that ultimately impinge on locomotor rhythm from the web (Fig. 6a and Supplementary Fig. 10). Within this network, there appears to be a major hub consisting of GSKT, which possess the greatest number of connections with other kinases and clock proteins (Fig. 6b). GSKT sends out regulatory outputs targeting DSOR1 and 4 components of the clockwork (Fig. 6a, b). This implicates that GSKT is a potentially essential regulator of the kinase network.

**GSKT acts to decrease TIM protein levels.** Given the central role of GSKT within the circadian kinase network, we further investigated how it affects locomotor rhythm by examining its predicted substrates (Fig. 7a). Among the potential substrates, SLMB, CUL-3, SGG, S6KII, and NEJIRE (NEJ) are already known to regulate the molecular clock and locomotor rhythm[6,40,41]. Under DD conditions, SLMB and CUL-3 exert effects on the clock by impinging on PER and TIM protein stability, whereas SGG times

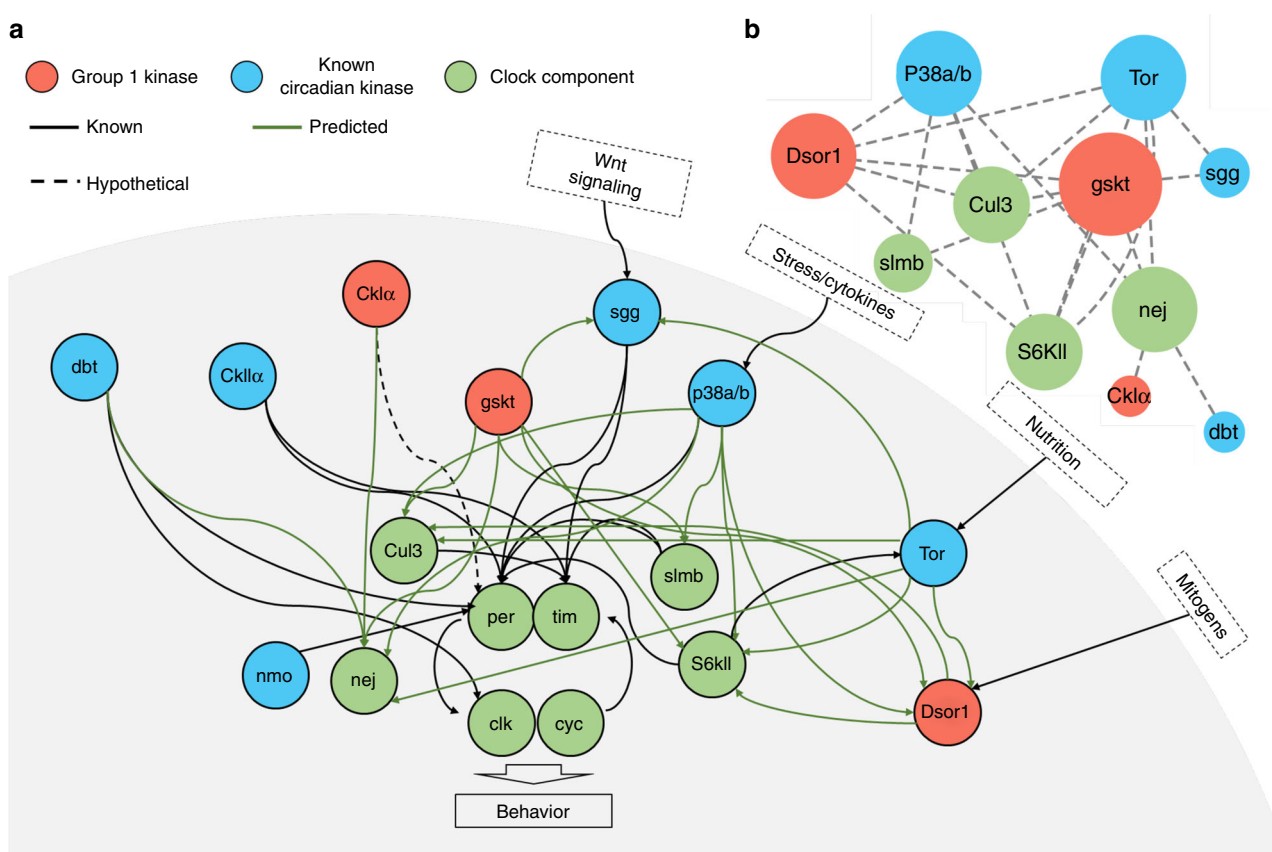

**Fig. 6 A kinase network that controls locomotor rhythm. a** A network of signaling cascades that regulate locomotor rhythms. Known regulations are indicated by black arrows, whereas predicted regulations are indicated by green arrows. Hypothesized regulations are indicated by dashed arrows. **b** Major hubs of the kinase network in **a**. Circle sizes are proportional to the numbers of interacting partners.

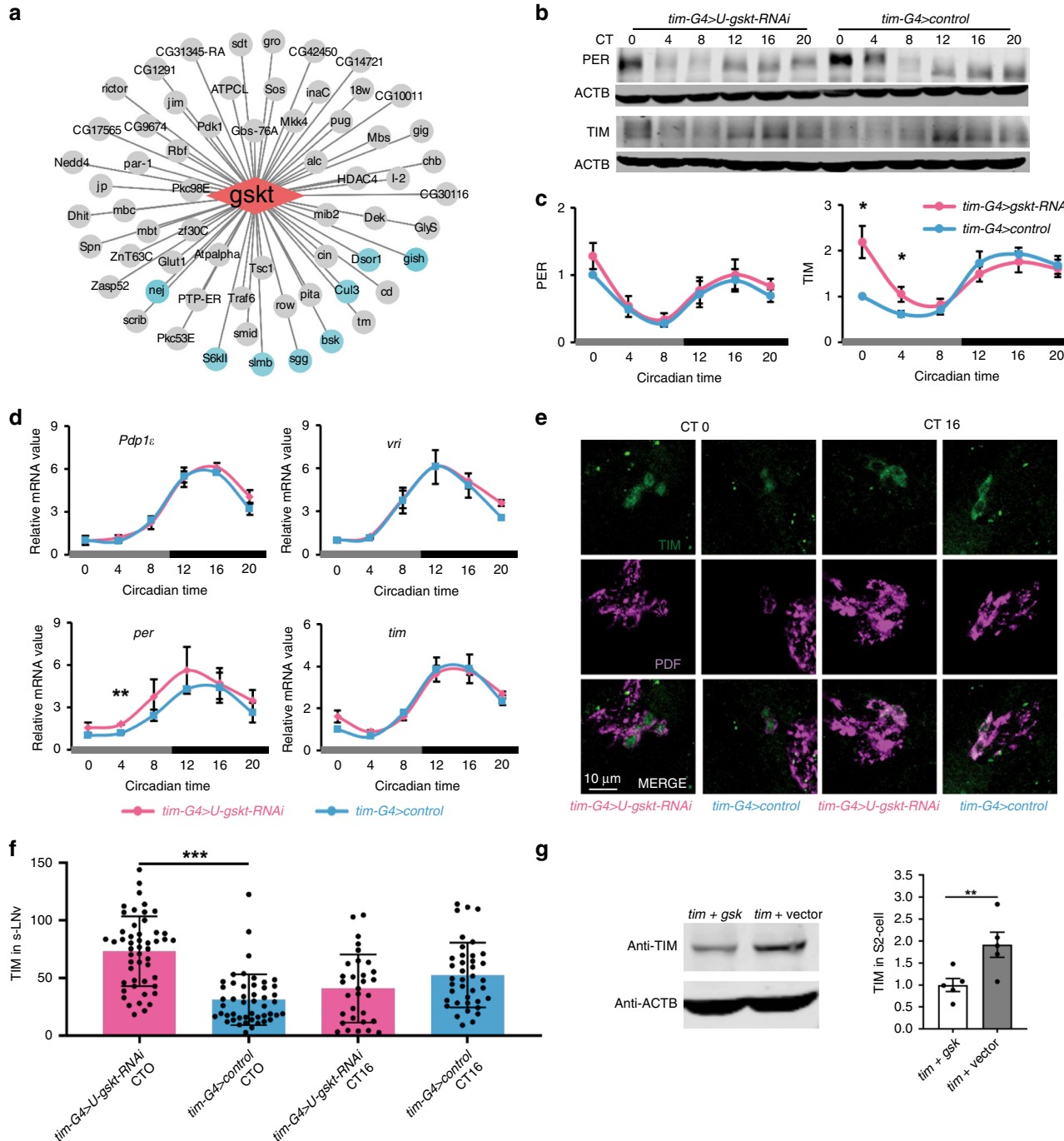

**Fig. 7 Knocking down *gskt* significantly increases TIM protein level. a** Potential substrates of GSKT filtered by PPI. Substrates that are involved (or potentially involved) in circadian regulation are labeled in green. **b** Western blots of proteins from whole-head extracts of *gskt-RNAi* (*timG4/+; Udcr2/Ugskt-RNAi*) and control (*timG4/+; Udcr2/+*) collected at the indicated circadian time. ACTB was used as a loading control. Repeated seven times independently with similar results. **c** Quantification of PER and TIM protein levels of blots in **b** (*n* = 7). PER and TIM protein levels were normalized to that of ACTB. For each time series, the value of the control at CT0 was set to 1. Two-tailed Student's *t* test, CT0, *\*p* value = 0.0154, CT4, *\*p* value = 0.0395. **d** Plots of relative mRNA abundance vs. circadian time for clock genes determined by qRT-PCR in whole-head extracts of *gskt-RNAi* and control flies collected on DD1 (*n* = 3). For each time series, the value of the control at CT0 was set to 1. Two-tailed Student's *t* test, *\*\*p* value = 0.0025. **e** Brains from *gskt-RNAi* and control flies collected at indicated circadian time and were immunostained with TIM (green) and PDF (red) antisera. The scale bar represents 10 μm. **f** Quantification of TIM protein levels in the s-LNvs of images in **e** (From left to right: *n* = 50, 48, 31, and 39; one-way ANOVA, CT0 *p* value = 6.84042 × 10$^{-12}$, CT16 *p* value = 0.09995). **g** Left panel: western blots of proteins from S2 cells co-transfected with pActin-*tim* and pActin-*gskt* or empty vector. Repeated five times independently with similar results. Right panel: quantification of TIM protein levels from the blots on the left (*n* = 5, two-tailed Student's *t* test, *\*\*p* value = 0.0089). ACTB was used as a loading control. Owing to large differences in molecular weight, ACTB was not ran on the same blot with PER/TIM although they were from the same sample. Error bars represent SEM. G4, GAL4; U, UAS. Source data are provided as a Source Data file.

the clock by regulating PER/TIM nuclear translocation[25,41,42]. S6KII is believed to regulate the clock by targeting SGG and possibly PER, whereas NEJ binds with CLK/CYC to modulate transcription of clock genes[40,43]. Therefore, we first tested whether knocking down *gskt* affects PER and TIM protein level in fly heads. Indeed, we observed significantly elevated TIM protein level and a trend of increase in PER protein level when *gskt* is knocked down (Fig. 7b, c). Meanwhile, *tim, vri,* and *Pdp1ε* mRNA levels are not significantly altered, whereas *per* mRNA level is moderately increased (Fig. 7d). As the period of locomotor rhythm in DD is determined by the small ventral lateral neurons (s-LNvs)[44], we assessed TIM levels in these cells and consistent with the changes observed at whole-head level, TIM is significantly increased at CT0 (Fig. 7e, f). Moreover, we co-expressed GSKT and TIM in *Drosophila* S2 cells and found significant reduction of TIM level compared to the control (Fig. 7g). Taken together, these results indicate that GSKT acts to reduce TIM protein level via post-transcriptional mechanism. Because among the five potential substrates that may mediate the effects of GSKT on the clock, only SLMB and CUL-3 are known to affect TIM protein level, this provides some potential candidate mechanisms regarding how GSKT may regulate TIM[41]. It is noteworthy that among the predicted substrates of GSKT are DSOR1, BSK, and GISH, implicating these kinases may function together with GSKT to regulate locomotor rhythms.

In short, this demonstrates an example of how our predictions can aid the process of characterizing the molecular actions of a candidate kinase in the circadian system.

## Discussion

Here in our phosphoproteomic study, we detected 14,946 non-redundant p-sites, including 9333 sites that have not been previously reported, which demonstrates the sensitivity of our system and provides a rich resource for future research. To further ensure the reliability of our data set, we removed all mRNAs with FPKM < 1 and their corresponding peptides/phosphopeptides, which has not been conducted in previous circadian phosphoproteomic studies[3–5]. In an attempt to optimize our iCMod pipeline, we removed this filter and re-performed the analysis only using the proteomic and phosphoproteomic data (Supplementary Fig. 11). We found this considerably lowered the accuracy of circadian kinase prediction. Moreover, we only analyzed mRNAs, peptides, and p-sites that can be detected at all time points rather than just some of the time points. Most importantly, we normalized the raw phosphorylation abundance of each p-site to its raw protein abundance to acquire NCPs, whereas previous studies directly analyzed phosphoproteomic data to identify oscillating p-sites[3,5]. These additional processing (especially the normalization step) makes a tremendous difference in the cycling p-sites identified (only 36% overlap), indicating the necessity of conducting at least the normalization procedure as this may completely change the oscillatory phosphorylation landscape identified. Wang et al.[4] focused on cycling p-sites of non-rhythmic proteins, but this will miss out on the cycling p-sites of rhythmic proteins. It is possible that both phosphorylation and protein levels are rhythmically regulated but their phases are different. Indeed, we observed that ~29% of the proteome show a negative correlation of its temporal expression profile with corresponding phosphorylation level.

We took one-step further and tested whether these global molecular oscillations are regulated by the molecular clock. Indeed, we found that the majority of the oscillations are damped in *per01* flies, suggesting that they are driven by the clock. Robles et al.[3] found that the amplitude of oscillation at phosphorylation level is much higher than at transcript or protein level.

Consistently, we have also observed this trend in our data with the amplitude of the phosphoproteome slightly higher than that of the proteome. However, we observed much lower cycling amplitude of global phosphorylation compared to what has been reported by Robles et al.[3] This is in part owing to differences in assessment method. Robles et al. adopted label-free quantification, whereas we used TMT labeling, which is known to compress differences in intensity levels[45]. Another possibility is that cells of the fly head are more heterogeneous than that of the mouse liver, thus the rhythms are less synchronized and robust. In addition, the molecular clock is not present in all cells in the head[46]. Therefore, oscillatory expression patterns in the clock cells could be masked by constant expression patterns in non-clock cells. Moreover, the amplitude that we calculated has been normalized to the protein level. This could be another reason for not observing highly robust cycling of the phosphoproteome. Interestingly, NCPs are enriched in proteins that function at the synapse and/or in the axon, which indicates that these sites favor rhythmic phosphorylation as a major contributor to rhythmicity in synaptic and axonal processes. This makes sense because neurites and in particular synapses are where most of the neural activities occur and thus are in high demand for energy. Cycling of phosphorylation is a much more economical way to bring about rhythmicity in protein function compared with cycling of protein level[47]. It is noteworthy that we observe substantial oscillations at mRNA, protein and phosphorylation levels even in *per0* animals. This is not unprecedented as Hughes et al.[48] have also reported 59 cycling mRNA in *per0* flies under DD (JTK cycle, *p* value < 0.01), which is ~1/4 of the number of cycling transcripts in WT. We observe a comparable ratio here. Interestingly, we observe even more oscillations at protein and phosphorylation levels. It is possible that some of these rhythms are residual light-driven oscillations from LD cycle. They may also reflect the activities of some unknown PER-independent oscillator(s) that is masked in the presence of PER. Further studies are required to characterize the nature of these oscillations.

Among the 27 predicted circadian kinases, 13 (48%) are involved (or potentially involved) in rhythm regulation. It should be noted that one of our identified circadian kinases, CkIalpha, has very recently been reported to interact with DOUBLETIME (DBT) for synergistically regulating PER in fly brain[49]. This finding further supports the accuracy of our predictions. As for the remaining 14 kinases that we have not been able to demonstrate a clear role in locomotor rhythm regulation, they may participate in modulating rhythms of other biological processes such as metabolism or sensory functions. 12% of cyclic proteins contain NCPs that are predicted to be phosphorylated by the 10 circadian kinases (including the seven known circadian kinases and three Group 1 kinases), which suggests these sites to be potential targets of these kinases. The cyclic phosphorylation on these proteins could result in cycling of protein stability and thus circadian oscillation in their abundance. In all, 69% of the cyclic proteins do not contain NCPs but are predicted to interact with at least one protein that contains NCP(s) potentially modified by one or more of the 10 kinases. These interactions may confer rhythmic modulation, rendering oscillatory protein levels. A previous study reported that CLK binds to ~800 genomic sites in a temporal-dependent manner and 267 of these CLK direct target sites show oscillating occupancy of RNA polymerase II, suggestive of rhythmic transcription from these sites[50]. However, there are still many cyclic mRNAs that do not appear to be direct targets of CLK. Here besides CLK, we predicted additional 32 transcription factors to participate in activating the transcription of the cyclic mRNAs that we detected here, which means these transcriptional activators could function together with CLK to generate rhythms in the transcriptome. Moreover, 30 of the transcription factors

are predicted to be phosphorylated by the 10 kinases, which could result in circadian cycling of their activities in transcriptional regulation. Taken together, our findings imply a role for these circadian kinases not only in generating rhythmic phosphoproteome, but may also contribute to global oscillations of the proteome and transcriptome. Clearly much more needs to be done to validate the actions of these kinases, which is beyond the scope of this study. Nonetheless, we propose a complex web of interactions based on our predictions in conjunction with literature that integrates these kinases into multi-omics circadian regulation.

We extracted and refined a small subset of the signaling web that reveals how the 10 kinases form a network to influence locomotor rhythm. It is surprising to find GSKT as a potential key regulator of the kinase network. *gskt* has been reported to be expressed in male germline cells in flies and required for male germline survival[28]. Interestingly, recent RNA-seq analysis revealed expression of *gskt* in circadian neurons[51] and moreover *gskt* was identified to be rhythmically translated based on ribosomal profiling of fly heads[24], implicating a potential role of GSKT in circadian regulation. Consistently, here we demonstrate expression of *gskt* in the head and a role for timing locomotor rhythm. Although GSKT is a paralog of SGG and deficiency in either one of these two kinases leads to lengthened period, they appear to have taken on different functions within the circadian kinome based on our predictions. Indeed, our experimental results demonstrate a role for GSKT in reducing TIM protein level, whereas SGG is known to phosphorylate PER/TIM and promotes their nuclear entry but does not appear to affect their protein levels[25,42,52]. Further studies are required to verify the kinase–substrate relationships in the circadian kinase network, but nevertheless, we believe this network provides a basic framework regarding phosphorylation regulation of the molecular clock that control locomotor rhythm.

In conclusion, our data reveal a circadian kinome that is potentially responsible for global molecular oscillations. We propose an intricate web formed by these kinases and their substrates, which in our opinion, is a significant step forward in understanding phosphorylation regulation of the clockwork from a system's level.

## Methods

**Fly stocks**. For RNA-seq, proteomics, and phosphoproteomics studies, male $w^{1118}$ (Bloomington Stock Center, BL3605) flies were crossed with $y^1 w^*$ and $per^{0153}$, respectively. Male flies of $F_1$ generation were used for omics study. For behavioral experiments, only male flies were used. Fly lines used are listed in Supplementary Table 1. We backcrossed $gish^{EY06451}$, $hep^{G0107}$, $Asator^{KG05051}$ lines onto the isogenic $w^{1118}$ background for two generations. For Western blotting, quantitative real-time PCR (qRT-PCR) and immunofluorescence analysis, both male and female progenies were used. *UAS-gskt-RNAi* (line1, V25640) was used for molecular analysis and GD background control was used for control.

**Fly head collection**. Flies were collected within 7 days of eclosion and entrained in 12 h/12 h light–dark (LD) schedule for 3 days. After that, flies were transferred into DD. On the first day of DD, flies were collected and frozen at 3 h intervals for WT flies (WT_0, WT_3, WT_6, WT_9, WT_12, WT_15, WT_18, and WT_21) and $per^0$ flies ($per^0$_0, $per^0$_3, $per^0$_6, $per^0$_9, $per^0$_12, $per^0$_15, $per^0$_18, and $per^0$_21). Frozen flies were vortexed for 10 s to separate the head from the body. Fly heads were collected and stored at −80 °C.

**RNA extraction and RNA-seq**. In all, 200 fly heads were homogenized in Total RNA Isolation (TRIzol) Reagent (Invitrogen), by using a handheld motor with plastic pestle. After mixing with trichloromethane, homogenates were centrifuged at 12,000 × *g* and suspension was precipitated with 75% ethanol. After air dry, total RNA was treated with RQ1 DNase (Promega) to remove genomic DNA and stored at −80 °C.

Before sequencing, RNA purity was measured by NanoPhotometer (IMPLEN, CA, USA), whereas the concentration and integrity of RNA were assessed by Qubit RNA Assay Kit in Qubit 2.0 Flurometer (Life Technologies, CA, USA) and RNA Nano 6000 Assay Kit of the Agilent Bioanalyzer 2100 system (Agilent

Technologies, CA, USA) (Supplementary Data 9). Agarose gel electrophoresis was also conducted for all samples (Supplementary Fig. 12a). According to the quality control, all 32 samples were classified as Class A, which represents the highest quality and RNA-seq profilings could be conducted more than twice for each sample. To construct the libraries for mRNA sequencing, NEBNext Ultra RNA Library Prep Kit for Illumina (NEB, USA) was used following the manufacturer's recommendations using 1.5 μg RNA per sample. Poly-T oligo-attached magnetic beads were employed to purify mRNA from total RNA, and then fragmentation was carried out with divalent cations under elevated temperature in NEBNext First Strand Synthesis Reaction Buffer (5×). First and second strand complementary DNAs (cDNAs) were synthesized by using random hexamer primers. For hybridization, NEBNext Adaptor with hairpin loop structure was ligated onto the 3' ends of the fragments, which were then purified by AMPure XP system (Beckman Coulter, Beverly, USA) to control for the length. PCR was performed after treating the cDNA with USER Enzyme for 15 min at 37 °C followed by 95 °C for 5 min. The purity of PCR products and assessment of library quality were performed on the Agilent Bioanalyzer 2100 system. In order to perform cluster generation of the index-coded samples, HiSeq 4000 PE Cluster Kit (Illumia) was used on a cBot Cluster Generation System. Sequencing of library preparations was performed on Illumina Hiseq 4000 platform.

**Protein extraction for proteomic analysis**. Fly heads were homogenized by sonication for 5 min in urea lysis buffer (8 M urea, 1× proteinase inhibitor and phosphatase inhibitor (Roche), 2 mM EDTA). Homogenates were centrifuged at 20,000 × *g* for 10 min and supernatants were collected. Finally, protein concentration was measured by BCA Protein Assay Kit (Thermo Scientific) and adjusted to be consistent across the time series (Supplementary Data 9). According to the SDS-PAGE results (Supplementary Fig. 12b), all 33 samples were classified as Class A, which represents the highest quality and further experiments could be performed more than twice for each sample.

**Isolation of peptides and TMT labeling**. In order to digest the proteins, protein solution was first treated by 5 mM dithiothreitol (DTT) at 56 °C for 30 min, followed by alkylation with 11 mM iodoacetamide for 15 min at room temperature in the dark. After that, 100 mM triethylammonium bicarbonate (TEAB) was used to dilute the protein samples to reduce the concentration of urea to <2 M. Two trypsin digestions were performed, using the mass ratio of 1:50 trypsin-to-protein for overnight treatment and 1:100 for 4 h, respectively.

After digestion, Strata X C18 SPE column (Phenomenex) and vacuum-dry were used to desalt the peptides and then the peptides were reconstituted with 0.5 M TEAB. The peptides were subsequently processed by TMT kit following the manufacturer's recommendations. The peptides were incubated with labeling reagent at room temperature for 2 h, followed by desalting and vacuum drying.

**Phosphopeptide enrichment**. High pH reverse-phase HPLC with Thermo Betasil C18 column was used to fractionate the tryptic peptides into fractions. The peptides were first separated into 60 fractions by a gradient of 8–32% acetonitrile (pH 9.0) for 60 min and then pooled into eight fractions. These fractions were subsequently vacuum dried.

To enrich phosphopeptides, Ti$^{4+}$-immobilized metal affinity chromatography microsphere suspension with vibration in loading buffer (50% acetonitrile and 6% trifluoroacetic acid) was used to incubate the peptide mixtures. The IMAC microspheres with enriched phosphopeptides were then collected by centrifugation. In order to remove the nonspecifically adsorbed peptides, 50% acetonitrile with 6% trifluoroacetic acid and 30% acetonitrile with 0.1% trifluoroacetic acid were used to wash the microsphere successively. After eluting the enriched phosphopeptides by vibration with elution buffer containing 10% NH$_4$OH, the supernatant containing phosphopeptides was collected and lyophilized for further analysis.

**LC-MS/MS analysis**. Liquid phase A (0.1% formic acid) was used to dissolved the tryptic peptides, which were then loaded onto a home-made reversed-phase analytical column (length: 15 cm, i.d.: 75 μm) and separated by EASY-nLC 1000 ultraperformance liquid chromatography system. Liquid phase B contains 0.1% formic acid in 98% acetonitrile. Liquid phase gradient setting was as follows: 0~50 min, 5–25% B; 50~62 min, 25–38% B; 62~66 min, 38–80% B; 66~70 min, 80% B. The flow rate was maintained at 400 nL/min.

Peptides were subjected to NSI ion source for ionization followed by tandem mass spectrometry (MS/MS) in Q ExactiveTM Plus (Thermo). The electrospray voltage was set to 2.0 kV and Orbitrap was used for detection and analysis. For primary MS, the scan range was 350–1800 m/z at a resolution of 70,000. Subsequently, normalized collision energy was set at 28% for selected peptides undergoing secondary MS/MS, with scan range starting at 100 m/z and resolution set at 17,500. A data-dependent procedure that alternated between one MS scan followed by 20 MS/MS scans was applied with 15.0 s dynamic exclusion. Automatic gain control (AGC) was set at 5E4.

**Standard database search**. MaxQuant (v.1.5.3.30)[54] was used for standard database search of MS/MS raw data. MS/MS spectra was searched against

*Drosophila* proteome database obtained from UniProt (Version 201706)[18], which contained 13,558 unique fly protein sequences. The digestion mode was set to Specific and Trypsin/P was chosen as cleavage enzyme allowing up to two missing cleavages. Carbamidomethyl (C) was the fixed modification, while Oxidation (M) and Acetyl (Protein N-term) were the variable modifications for searching both proteomes and phosphoproteomes, and Phospho (STY) for phosphoproteomes only. Seven was set as the minimum peptide length and 4600 Da as the Maximum peptide mass. The false discovery rates for peptide-spectrum match, protein and p-site decoy fraction were all set to <1% and minimum score for modified peptides was set to >40.

**Locomotor activity monitoring**. To knock down or overexpress a kinase, tim-GAL4;UAS-*dcr2* and UAS-*dcr2;cry*-GAL4-16[55] were crossed to RNAi lines and overexpression lines. For controls, UAS and GAL4 lines were crossed to the genetic background controls for TRIP, GD, and KK RNAi collections, *w1118* and *yw* strains. Flies were reared on standard cornmeal–yeast–sucrose medium and kept in LD cycles at 25 °C. Locomotor activity levels of adult male flies were monitored by *Drosophila* Activity Monitoring System (DAMS, TriKinetics) for 7 days of LD followed by 7 days of DD.

**PDF and TIM immunofluorescence and microscopy**. Adult male flies were entrained for 3 days at 25 °C and anesthetized with $CO_2$. Brains were dissected in PBS buffer containing 3.7% formaldehyde. After fixation at room temperature for 30 min, the brains were rinsed two times in PBS and incubated in PBS with 1% Triton for 10 min at room temperature. The brains were then incubated in 5% donkey serum diluted in PBT (PBS with 0.5% Triton) for 30 min at room temperature, followed by incubation for two days in a mixture of 1:100 rat anti-TIM (rat TIM antibody is a generous gift from Dr. Joanna Chiu) and 1:50 mouse anti-PDF (DHSB) in PBT containing 5% donkey serum at 4 °C. After PBT rinses for six times, the brains were incubated with 1:500 donkey anti-rat AlexaFluor 488 (Invitrogen) for TIM immunostaining and 1:10–1:20 donkey anti-mouse Alexa-Fluor 594 (Invitrogen) for PDF immunostaining in PBT overnight at 4 °C. After final rinses in PBT, brains were mounted in 80% glycerol diluted in PBS. PDF/TIM-labeled specimens were photographed with ×60 oil lens by Olympus FV1000 laser scanning confocal microscope (Olympus). The microscope, laser, and filter settings for a given experiment were held constant.

**Transient transfection**. S2 cells were plated in 12-well plates and transfected with FuGENE 6 (Promega). DNA plasmids used for transfections were as follows: pActin-HA-tim-V5, pActin-gskt-V5-6×His and pActin-V5-6×His. Cells were harvested 44 h after transfection.

**Protein extraction and western blot**. Proteins were extracted from fly brains or S2 cells using SDS lysis buffer (10 mM Tris-base, 1 mM sodium orthovanadate, 1% SDS, pH 8.0, 1 mM DTT, 1× proteinase inhibitor and phosphatase inhibitor (Roche)). After homogenization, protein lysates were centrifuged at $12,000 \times g$ for 15 min at 4 °C and incubated at 95 °C in loading buffer for 5 min. Equal amounts of protein were loaded into each well on 5% or 8% SDS-PAGE gels and then transferred to nitrocellulose membranes for 2 h at 90 V. Membranes were incubated with primary antibody at 4 °C overnight followed by secondary antibody at room temperature for 1 h. The primary antibodies used were as follows: guinea pig PER (1:1000), rat TIM (1:1000), rabbit ACTB (1:5000, ABclonal), guinea pig SGG (1:1000), and phospho-SGG Y214 (1:1000, Abcam). Donkey secondary antibodies (1:10000 dilution) were conjugated either with IRDye 680 or IRDye 800 (LI-COR Biosciences) and visualized with an Odyssey Infrared Imaging System (LI-COR Biosciences). Rat TIM antibody, guinea pig SGG antibody, and guinea pig PER antibody were generous gifts from Dr. Joanna Chiu.

**PCR and qRT-PCR**. PCR was performed with Taq Plus MasterMix (CWbiotech). The PCR reaction was performed as following: 94 °C for 2 min followed by 94 °C for 10 s, 57 °C for 15 s and 72 °C for 1 min 20 s for 35 cycles. Quantitative real-time PCR was performed with One-Step RT-PCR SuperMix (Transgen). The PCR reaction was performed as follows: 45 °C for 5 min; 94 °C for 2 min; 94 °C for 5 s, 58 °C for 15 s, 72 °C for 20 s for 40 cycles (Applied Biosystems). The $^{\Delta\Delta}$CT method was used for quantification. *Beta-Actin* was used as internal control. The primers used are listed in Supplementary Table 2.

**RNA-seq analysis**. For the analysis of the RNA-seq data, raw reads were first mapped to the reference genome of *D. melanogaster*, which was downloaded from Ensembl (release version 90, http://www.ensembl.rog/)[56]. The software packages of Bowtie2 (version 2.2.4) and TopHat (version 2.1.1) were used to generate the BAM files, whereas Cufflinks (version 2.2.1) was employed to assemble the reads and calculate the expression levels of individual mRNAs based on FPKM values[57].

**Sample correction and normalization**. Unexpectedly, six samples (WT_0, WT_6, WT_12, WT_18, *per0*_6, *per0*_18) failed to pass the quality tests before RNA-seq in the first round. Therefore, we re-prepared these samples by adding WT_3 as a normalization control, and re-performed the transcriptomic, proteomic, and

phosphoproteomic analysis. Owing to the limitation that only 10 samples can be simultaneously labeled and analyzed by the TMT technology, four batches of LC-MS/MS analyses were carried out for all 32 samples (Supplementary Table 3).

We observed a considerable fluctuation between Batch 4 and the other three batches. To improve the consistency of the four batches, we used the WT_3 sample quantified in both Batch 1 and Batch 4 as the normalization control for proteomic and phosphoproteomic data. First, all proteins and p-sites quantified in both batches for the WT_3 sample were picked out, whereas the Batch 1: Batch 4 ratio was calculated for each protein and p-site, respectively. Then all intensity values of these proteins and p-sites quantified in other six samples of Batch 4 were normalized with the ratio.

**Customized reference databases**. For each batch, only mRNAs with FPKM ≥ 1 in at least one sample were considered, and their corresponding proteins were used to construct a sample-specific reference database. In total, there were 9461, 9619, 8649, and 8610 non-redundant protein sequences reserved for Batch 1, 2, 3, and 4, respectively. For these fly proteins, their reverse decoy sequences were separately generated for each database. Then MaxQuant (v.1.5.3.30)[54] was used for searching each reference database to identify peptides and phosphopeptides from proteomic and phosphoproteomic MS/MS spectra, respectively, with identical parameters in standard database search.

**Proteomic and phosphoproteomic data normalization**. Sample-based normalization was conducted for the raw proteomic and phosphoproteomic data using the global centering (GC) method[58]. For each sample, the identified non-phosphorylated peptides and phosphopeptides were re-mapped to 13,558 non-redundant fly protein sequences of the *Drosophila* proteome set, and the average intensity value of all proteins and p-sites was normalized to 1 (Mean = 1) for proteomic and phosphoproteomic data sets, respectively. To exclude the bias of GC normalization for the correlation analysis, we employed four additional methods for normalization of the proteomic data using metaX (http://metax.genomics.cn/)[59].

**Computational identification of circadian oscillations**. Circadian oscillations at different levels were identified by MetaCycle, an integrative R package that incorporated three computational programs including ARSER, JTK_CYCLE, and Lomb-Scargle[60]. All three methods were tested, and very few hits were detected by JTK_CYCLE and Lomb-Scargle. Only ARSER recognized a considerable number of mRNAs, proteins and p-sites to be potentially rhythmic ($p$ value < 0.01).

**Prediction of ssKSRs with GPS**. Previously, we developed a software package of GPS (http://gps.biocuckoo.org/)[22], which classifies protein kinases into a hierarchical structure at four levels, including group, family, subfamily, and single kinase. In total, GPS contains 144 and 69 individual predictors to predict ssKSRs from primary sequences of proteins for serine/threonine kinases (STKs) and tyrosine kinases (TKs), respectively. From GPS 2.1, we manually selected 48 and 15 predictors for 153 STKs and 16 TKs in *D. melanogaster*, respectively. To increase the coverage of p-sites with predicted protein kinases, a low threshold was chosen with a false positive rate (FPR) of 10% for STKs and 15% for TKs, respectively. GPS 2.1 was then used to predict ssKSRs for all 789 p-sites identified by iCMod in all 16 samples of WT flies.

**Two-sided hypergeometric test**. In iCMod, two-sided hypergeometric test was adopted for identification of potential circadian kinases, which significantly prefer to modify NCPs rather than non-oscillated p-sites based on predicted ssKSRs. For each protein kinase $k_i$ ($i = 1, 2, …, 169$), we defined the following:

$N$ = number of p-sites identified by iCMod.
$n$ = number of iCMod p-sites predicted to be phosphorylated by $k_i$.
$M$ = number of NCPs.
$m$ = number of NCPs predicted to be phosphorylated by $k_i$.

The enrichment ratio (E-ratio) of $k_i$ was computed, and the $p$ value was calculated with two-sided hypergeometric distribution as below:

$$\text{E} - \text{ratio} = \frac{m}{M} / \frac{n}{N} \tag{1}$$

$$p - value = \sum_{m'=m}^{n} \frac{\binom{M}{m'}\binom{N-M}{n-m'}}{\binom{N}{n}} (E - \text{ratio} \geq 1), \tag{2}$$

$$p - value = \sum_{m'=0}^{m} \frac{\binom{M}{m'}\binom{N-M}{n-m'}}{\binom{N}{n}} (E - \text{ratio} < 1). \tag{3}$$

All protein kinases with substrates significantly enriched in NCPs ($p$ value < 0.05 and E-ratio > 1) were referred to as potential circadian kinases.

Two-sided hypergeometric test was also adopted for the GO-based enrichment analyses at mRNA, protein and phosphorylation levels. GO annotation files (released on 22 January 2018)[61] were downloaded from the EBI Web site (https://www.ebi.ac.uk/QuickGO/) and contained 12,452 fly proteins with at least one GO term.

We mapped 2768 genes with rhythmic mRNA expression collected in CGDB[23] and 1255 translationally rhythmic genes identified by translating ribosome affinity purification[24] to our cycling mRNA, protein and phosphoprotein data sets identified by iCMod. Two-sided hypergeometric test was performed for the enrichment analysis.

**Re-construction of the circadian kinase signal web**. At the phosphorylation level, a kinase–substrate network was first determined from predicted ssKSRs of NCPs regulated by the seven known circadian kinases and 3 Group 1 kinases. At the protein level, experimentally validated and pre-calculated PPIs from six public databases including BioGRID[62], DIP[63], MINT[64], I2D[65], IntAct[66], and STRING[67], with a total of 2,280,705 PPIs in 18,308 fly proteins, were downloaded and integrated into the kinase–substrate network of the 10 kinases. A cyclic protein identified by iCMod that interacts with at least one member in the kinase–substrate network was retained and incorporated into the signal web. At the mRNA level, annotated transcription factors in *D. melanogaster* were downloaded from a previously developed database AnimalTFDB 3.0[68] to predict transcription factors that activate the transcription of genes that oscillate at mRNA level (http://bioinfo.life.hust.edu.cn/AnimalTFDB/). For the prediction of transcription factors based on the presence of transcription factor binding sites (TFBSs), potential transcription factor binding regions (2000 bp upstream and 500 bp downstream) of genes with circadian mRNAs were extracted from the.gtf file downloaded from Ensembl[56], then the TFBS predictor implemented in AnimalTFDB 3.0[68] was used. Because AnimalTFDB 3.0[68] could only predict TFBSs for human transcription factors, their orthologs in *D. melanogaster* were computationally determined through a classical approach of reciprocal best hits (RBHs)[69]. Then, we searched a comprehensive phosphorylation database EPSD (http://epsd.biocuckoo.org/) and retained transcription factors with at least one experimentally identified p-site. GPS 2.1[22] was adopted to predict potential ssKSRs for these mapped p-sites, and PPIs between kinases and transcription factors were used to reduce the false positive hits. Only transcription factors with at least one ssKSR regulated by one or more of the 10 kinases were incorporated into the web. In addition, we incorporated 2092 and 1141 TFBSs of CLK and CYC, respectively, from published data set using a cutoff of read density ≥ 2, as well as kinase–substrate relations published in literature[39,70].

To model the kinase network that regulates locomotor rhythm, predicted ssKSRs between the 10 kinases and genes known to function in the clockwork were retained. After filtering by PPIs between kinases and substrates, the core network contains the 10 kinases, four core clock genes (PER, TIM, CLK, and CYC) and three genes known to regulate the clock (S6KII, SLMB, and CUL-3).

**Circadian analysis of locomotor activity**. For DD rhythmicity, chi-squared periodogram analyses were performed by Clocklab (Actimetrics, Wilmette, IL). Rhythmic flies were defined as those in which the chi-squared power was ≥ 10 above the significance line. Period calculations considered all flies with rhythmic power ≥ 10. Dead flies were defined by 0 activity on DD7 and removed from analysis.

**TIM immunofluorescence quantification**. For TIM intensity quantification, all slides were coded for sample identity and remained until the numerical analysis stage. The contour of each cell was circled and staining intensity was measured from single slice image using ImageJ (NIH).

**Reporting summary**. Further information on research design is available in the Nature Research Reporting Summary linked to this article.

## Data availability
The source data underlying Fig. 3h, i, 7b–d, f, and g and Supplementary Figs. 7–9 are provided as a Source Data file. The RNA-seq data have been deposited into NCBI Sequence Read Archive (SRA, [https://www.ncbi.nlm.nih.gov/sra]) with the data set identifier SRP145574. The raw mass spectrometry proteomics data have been deposited into the integrated proteome resources (iProX, [http://www.iprox.org/]) with the data set identifier IPX0001218000. The PPI data sets were integrated from six public databases including BioGRID ([https://thebiogrid.org/], downloaded in 09/2016), DIP ([https://dip.doe-mbi.ucla.edu/dip/Main.cgi], downloaded in 11/2016), MINT ([https://mint.bio.uniroma2.it/], downloaded in 10/2016), I2D ([http://ophid.utoronto.ca/ophidv2.204/], downloaded in 09/2015), IntAct ([https://www.ebi.ac.uk/intact/], downloaded in 10/2016), and STRING ([https://string-db.org/cgi/], v10, downloaded in 11/2016). Known p-sites were downloaded from eight public databases, including dbPAF ([http://dbpaf.biocuckoo.org/], downloaded in 01/2018), dbPTM 3.0 ([http://dbptm.mbc.nctu.edu.tw/], downloaded in 12/2015), Phospho.ELM ([http://phospho.elm.eu.org/], downloaded in 12/2015), PHOSIDA ([http://141.61.102.18/phosida/index.aspx], downloaded in 10/2015), PhosphoPep 2.0 ([http://www.phosphopep.org/], downloaded in 10/2015),

PhosphoSitePlus ([http://www.phosphosite.org/], downloaded in 09/2015), SysPTM 2.0 ([http://lifecenter.sgst.cn/SysPTM/], downloaded in 10/2015) and UniProt ([http://www.uniprot.org/], downloaded in 12/2015). GO annotation files (released on 22 January 2018) were downloaded from the EBI Web site (https://www.ebi.ac.uk/QuickGO/). Drosophila proteome databaseset was obtained from UniProt (Version 201706). Known circadian genes were downloaded from CGDB ([http://cgdb.biocuckoo.org/], downloaded 05/2018). Transcription factors in *D. melanogaster* were downloaded from AnimalTFDB 3.0 [http://bioinfo.life.hust.edu.cn/AnimalTFDB/], downloaded in 12/2018). The p-sites inof transcription factors were searched against the database EPSD ([http://epsd.biocuckoo.cn/], downloaded in 12/2018).

## Code availability
The source code of iCMod has been uploaded to GitHub [https://github.com/CuckooWang/iCMod].

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

## Acknowledgements

This work was supported by grants from the Natural Science Foundation of China (31930021, 31970633, 31671215, 31471125, 31671360, and 31801095), the Special Project on Precision Medicine under the National Key R&D Program (2017YFC0906600 and 2018YFC0910500), the National Program for Support of Top-Notch Young Professionals, Young Changjiang Scholars Program of China, the program for HUST Academic Frontier Youth Team, and the Fundamental Research Funds for the Central Universities (2019kfyRCPY043). We would like to thank Drs. Joanna Chiu (UC Davis), Yong Zhang (Univ. of Nevada, Reno), Ravi Allada (Northwestern University) and Guanghui Wang (Soochow University) for reagents used in this study. We thank Dr. Xi Zhou (Wuhan University) for *Drosophila* S2 cells used in this study. We thank Dr. Bing Zhang (Baylor College of Medicine) for his helpful comments. We would also like to thank Tsinghua Fly Center for providing fly stocks.

## Author contributions

Y.X. and L.Z. initiated the project and oversaw all aspects of the project. C.W.W carried out the data analysis and prediction. K.S. and S.S.M. performed the experiments. S.F.L., Y.Z., B.W., W.K.D., H.D.X., H.H., and A.Y.G. put forward helpful suggestions for the analysis of data. Y.X. and L.Z. wrote the manuscript with input from all the authors. All authors reviewed and approved the manuscript for publication.

## Competing interests

The authors declare no competing interests.
