## [Peer Review File · Nature Communications]

Reviewers' comments:

Reviewer #1 (Remarks to the Author):

In this manuscript, the authors proposed an integrative analysis of RNA-seq and LC-MS/MS data to identify circadian kinome in fruit fly head. Overall, this manuscript is good written and easy to follow. Through the computational analysis, the authors claimed that novel kinase regulators for circadian rhythm has been discovered. I have major concerns listed below.

1. The authors already found that the transcriptome-proteome correlation is extremely weak (1.79% positive, 1.53% negative). But authors persisted in using only high expressed mRNAs in the construction of reference database for MS/MS searching in the iCMod pipeline. I don't think this step is proper. It will generate too many false positives. The peptides and phosphopeptides identified may not expressed in the MS data.

2. Authors should provide evidence that they did good QC about the MS and RNA-Seq data. Since existing studies showed that correlation between mRNA and protein is generally ~ 0.40 (Vogel, C. and Marcotte, E.M, Nat Rev Genet 2012). The extremely low concordance between mRNA and protein expression suggesting there is something wrong with QC or library construction process. Inference based on unreliable data is questionable. PTM may explain some degree of the discordance, but the author should explain why their results are robust.

Minor:

3. There is a lack of experimental details on how they get the data for RNAi, using small interfering RNA (siRNA)?

4. I suggest to make the iCMod source code publicly available, so it can be tested.

Reviewer #2 (Remarks to the Author):

This is a potentially important manuscript. The authors have systematically studied with mass spectrometry circadian phosphorylation in head extracts. This was followed by identification of candidate kinases that might be responsible for these phosphorylation rhythms. Of the 20 candidates, 7 show circadian behavior phenotypes with RNAi-mediated knockdown (not including previously known circadian kinases). In a few cases, this is followed up by testing gene mutants or overexpressing wild-type or mutant proteins

The multi-omic nature of the paper could provide a very important resource for the circadian and fly community. The authors should therefore do everything they can to make these data easily accessible and searchable. It would however be important to get a better sense of the quality of the results. It is certainly reassuring that known circadian kinases were identified, but those tend to be quite broadly expressed, pleiotropic proteins (CKI, CKII, GSK-3 etc...). In particular, the authors should comment on whether they found in their data known phosphorylation rhythms, for example in PER, TIM or CLK. Also, if phospho-specific antibodies were available, it would be important to confirm a few of the new rhythmic phosphorylation sites (note that I am not expecting that the authors would generate such reagents for the present study).

The authors should be careful not to overstate their findings. At this point, the web of interactions they propose are not demonstrated in the context of circadian rhythms of phosphorylation. Of particular concern was the claim that GSKT is a "master regulator". There is no direct evidence for such a prominent role for GSKT at this point, only inference from previous studies performed in different contexts. Actually, the impact of GSKT seems quite mild on circadian rhythms (see below).

My last general concerns have to do with the functional studies (behavior, WB and IHC). It is curious that none of the 9 RNAi lines for BSK and none of the three mutants tested gave any phenotypes. I am concerned that the long period phenotype seen with overexpression of either WT or a dominant negative are overexpression artefact. The mutants on fig 5d give mild phenotypes. It would be important to ensure that these phenotypes really map to the relevant genes. The WB phenotypes seen with GSKT knockdown are rather weak, essentially affecting one time point for both PER and TIM. Error bars are larger on these time points, which make me concerned that the results might not hold under further examination. Also, the WB does not look right for PER, since PER phosphorylation is abnormal, with high phosphorylation levels at CT12 and low at CT0. It should be the other way around. The IHCs are not very convincing either. PDF signal is saturated and it is difficult to know where each sLNv is. In addition, according to Shafer et al (2002), TIM levels should be about the same at CT0 and CT 16, but with different localization. Here, in the control flies, TIM is very low at CT0, essentially invisible on the image provided. Actually, the experimental flies look more like what Shafer and al described.

Although I have listed some significant concerns, I would also re-emphasize my feeling that this could be a very important paper. The multi-omic approach is really interesting and the data should be very important, but they need to be validated more carefully and convincingly.

Specific points:

- 1) It is quite interesting that there is such a weak correlation between transcription and protein rhythm profiles. However, as mentioned above, it would be important to discuss validation of the data set. Were the expected rhythmic mRNAs and proteins detected, how do the present results compare to previous publications on the circadian transcriptome? Finally, does the rhythm in proteins fit with ribosomal profiling done by the Jackson's lab?
- 2) The authors indicate that 15% of protein or phosphorylation rhythms, as well as 7% of mRNA rhythms, are still present in *per0*. Is this indicative of the rate of false positive, which would then seem quite high? Previous studies have shown that all transcriptional rhythms are eliminated when core clock genes such as PER are absent. If these residual rhythms are real, where would they be coming from?
- 3) *cryGAL4-16* is not specifically expressed in clock neurons.
- 4) Page 12: where are the JNK1/2 phosphorylation sites residues in BMAL1? Are they present in CYC, which is quite small and contains essentially only the bHLH-PAS domains?
- 5) Page 14: another reason for low amplitude cycles could be that many tissues do not contain circadian clocks. Thus, in some tissues, a protein or its phosphorylation could be rhythmic, while in others it might not be. This also brings up a caveat of the approach, which should be discussed. Some cycling events could be missed because they are only present in a limited fraction of cells in the fly head.
- 6) Page 15, bottom. The presence of putative phosphorylation sites for a kinase does not demonstrate that the kinase phosphorylates these sites. The word "implicating" is thus too strong.
- 7) The identification of 32 putative transcription factors implicated in circadian transcriptional control is interesting and should be more prominently mentioned in the results section. Bring more attention to figure S7. However, I do not understand why CLK, CYC, VRI and PDP1 are missing from this list. In addition, again, the authors should refrain to say "computational analysis revealed that the transcription of all genes oscillating at mRNA level is activated by 32 transcription factors". This is way too strong. Computational analyses can only make predictions, which then need to be tested. Throughout the text, the authors will have to soften their words, unless they provide with direct experimental evidence.
- 8) It would be important to show that GSKT is expressed in circadian tissues. Maybe this has been shown in whole genome expression profiling of circadian neurons?
- 9) p.17. "we think" is not a good choice of words. In addition, there is nothing in the paper to indicate that expression of PER or TIM would be mediated by GSKT via SLMB or CUL3

Reviewer #3 (Remarks to the Author):

The authors systematically collected phosphoproteomics, proteomics and transcriptomics data from fly heads at different stages of the circadian cycle and used it to identify potential kinases responsible for the oscillatory phosphorylation patterns observed. There exist prior phosphoproteomics studies in the field including in other organisms, but they haven't extracted a kinase network from this data and most, to my knowledge, haven't normalized the data by proteome abundances, which is a very important step in getting the actual effects due to phosphorylation rather than proteome changes. They validated the role of some of the identified kinases in the circadian clock using knockdowns and identified GSKT as a master regulator of the kinase network.

Overall the work is well done and will be useful for the community. There are some issues that would be good to be addressed prior to publication.

Major issues

My major problem with the paper is the removal of phosphosites based on the mRNA expression of the respective gene. It is known and the authors themselves even show that mRNA levels are not correlated at all to protein or phosphoprotein abundances and given the different timelines of transcription and translation and degradation of the respective molecules, I believe it is not valid to remove data points based on the mRNA dataset. The authors should either prove that this is a valid thing to do or not do it at all. Examples of how to do this are:

Of 20 predicted kinases only 8 are validated. Without removing the mRNA data, do they still find the 8 that were validated? Of the 20, 7 are known. How many known ones did they miss, and would they have found them if they hadn't removed the data based on the mRNA levels. Additional evidence or arguments that this is a valid approach are needed before committing it to the public literature.

Finally, based on the location of GSKT in the network the authors claim it is a master regulator of the kinase network. I think that this is a bit of a leap. A master regulator means that it controls all the functions of the rest of the kinases, and this has not been shown in this study.

Minor issues

- Last sentence of abstract missing 'circadian' molecular landscapes.
- First sentence in Introduction doesn't make sense, must be some typo.
- Page 10- Top ten kinase families should be The top ten kinase families.
- Figure S7 can't really be read or interpreted, so it is best to also provide the information in an excel file where the pairwise relationships can also be read.

Response to Reviewers

To make it clear, we have used *italic* fonts for the reviewers' comments, **black** fonts for our replies, and **blue** fonts for revisions.

Reviewer #1 (Remarks to the Author):

This is an interesting concept with potential for leading to new applications and design of memristors but in my opinion it is not presented in a way that is convincing and therefore I recommend major revision.

We greatly thank the reviewer for confirming the novelty in the work, both from the concept and potential application perspectives. We also thank the reviewer for the suggestions/comments to strengthen the scientific presentation. Below please find our detailed responses to the reviewer's questions.

The structure of OmcS protein, that forms conductive filaments in Geobacter was recently published in Cell and is not mentioned. Indeed the molecular modeling used by the authors reported in supplementary figure 2 is based on pilA whose capacity to conduct electrons remains controversial. Furthermore, in the materials and methods section, synthesis and purification of protein nanowires, the authors report no attempt to unequivocally identify the molecular nature of the biological material used. For instance, MS would be a trivial measurement that would identify peptides that could be matched to proteins in the genome.

We appreciate the reviewer's update on the recent development in protein nanowires. We are well aware of the formation of OmcS filaments as one of the authors of our manuscript was also an author on the first publication of the OmcS filament structure (<https://www.biorxiv.org/content/10.1101/492645v1.abstract>). However, as recently reviewed in detail (*Front. Microbiol.* 2019, 10, 2078), the abundance of PilA-based filaments and OmcS-based filaments is highly dependent upon culture conditions. In the studies reported here the methods employed yielded filament preparations comprised primarily of PilA-based filaments. In contrast to the reviewer's comment, mass spectrometry (MS) is not a "trivial method" when identifying PilA-based filaments because the filaments are not denatured to subunits and the PilA subunit does not 'fly' in typical MS methods. However, it is possible to distinguish between PilA-based filaments and OmcS-based filaments based on the differences in their diameters. We demonstrate here that the diameters of the filaments employed were 3 nm, consistent with the diameter of PilA-based filaments and inconsistent with the 4 nm diameter of OmcS-based filaments (Fig. R1).

What is more relevant to this study is whether the coexistence of OmcS nanowire plays a role in the memristive effect. Therefore, we have done control study by using nanowires harvested from a strain of *Geobacter* in which the gene for OmcS was deleted. In the revised

manuscript, we demonstrate that protein nanowires obtained from a mutant strain of *Geobacter* that could not produce OmcS functioned as well in memristor devices (Fig. 2R) as protein nanowires derived from wild-type *Geobacter*. These results demonstrate that PilA-based filaments rather than OmcS-based filaments were responsible for memristor function.

It is also important to recognize that the co-existence of PilA-based and OmcS-based nanowires would not significantly alter the structure of the protein nanowire films.

For example, we have performed further molecular dynamics simulations by including OmcS nanowires. As both pilA and OmcS nanowires have helical structures, their combinations still yield molecular pores at the nanowire-nanowire interface (Fig. R3).

So in the revised manuscript, we have (1) added the possibility of the co-existence of OmcS nanowires by citing latest studies including the *Cell* paper, (2) extended the molecular dynamics simulations (e.g., adding Fig. R3 to Supplementary Fig. 3), and (3) discussed that OmcS nanowire is not likely to be the contributing factor, by adding the new results of Figs. R1 & 2 to a new Supplementary Fig. 1e and Supplementary Fig. 2) and the description in main text (page 4):

“...Two types of protein nanowires can be recovered from *Geobacter sulfurreducens*, protein wires that assemble from the pilin monomer PilA and wires that assemble from the c-type cytochrome OmcS.²⁸ The relative abundance of each type of wires depends upon the conditions under which the cells are grown. The protein nanowires in our preparations (Supplementary Fig. 1) had an average diameter of 2.9 ± 0.35 nm, which is consistent with the 3 nm diameter of pilin-based nanowires²⁸ and inconsistent with the 4 nm diameter of protein nanowires comprised of OmcS.^{30,31} Furthermore, as detailed below, devices constructed with protein nanowires harvested from a strain of *Geobacter sulfurreducens* in which the gene for OmcS was deleted yielded similar results (Supplementary Fig. 2). These results suggested that pilin-based protein nanowires were the important functional components.”

The use of acetone in one of the steps for fabrication of the memristor as reported in supplementary figure 3 leads me to question the structural integrity of the protein after the treatment and thus the relevance of the molecular modelling reported in figure 2.

The fact that as reported in the materials and methods section, fabrication of memristive devices, the authors report that drying at high temperature leads to material with properties similar to drying at room temperature again leads me to question the structural integrity of the biological material deposited, and I do not see in the description of the work evidence that the authors controlled this issue.

We thank the reviewer for the careful thought over the structural integrity in material after different treatments. Previous studies have shown that nanowires from *Geobacter* are stable in harsh environment (e.g., pH 2-10), against organic solvents (e.g., detergent), and at high temperature (e.g., 100 °C) (*Curr. Opin. Electrochem.* 2017, 4, 190; *mBio* 2017, 8, e00695-17). Such

Fig. R2. Sub-100 mV memristive switching in a device made from nanowires harvested from *Geobacter* with OmcS gene deletion.

Fig. R3. MD simulations of pilA-pilA, OmcS-OmcS, and OmcS-pilA nanowire interfaces (left), with corresponding range in the interfacial pore size (right).

stability is only found in some proteins, an analogous example is amyloid proteins. This can be a remind of nature's design to allow certain organisms to live in different environments (e.g., algae/bacteria in high-temperature environments (*Science* 1967, 158, 1012)).

To further address the reviewer's concern, we have done additional studies. First, the same nanowire film prepared at room temperature (RT) showed negligible change in conductance after it was heated to different temperatures of 50°, 70°, 90°, 110° and cooled down to RT (Fig. R4). Drop-casting acetone on the film yielded some conductance decrease but maintained the same level (yellow line). This conductance change can come from the sensing effect to acetone adsorption, which has been constantly observed in other thin-film devices/sensors (*Nanomaterials* 2017, 7, 339).

Fig. R4. Current-voltage curves (IVs) from the same protein-nanowire film after different treatments.

More importantly, we have also tested the memristive switching in three devices with the nanowires prepared by drying at (1) RT, (2) 110°, and (3) RT with acetone cleaning, respectively. The three devices showed consistent bio-voltage memristive switching (Fig. R5).

Fig. R5. Sub-100 mV switching in three devices with the nanowire film prepared in different conditions.

We believe that the prior studies and our new controls are convincing evidence that the nanowires can withstand the processing conditions for device application, although we acknowledge that current processes are still in the proof-of-concept demonstration stage. In the revised manuscript, we have cited the prior studies to indicate the stability in nanowires (page 8): “...The protein nanowires are stable under harsh chemical and temperature conditions, providing broad options for the incorporation in electronic devices.^{27,55}”

Figure 4 panel f: does the dashed line represent a mathematical model of the phenomenon or is it there as a visual guide. If it is the second option, it should be made of linear segments connecting the experimental points, as the authors used in panel c of figures 3 and 5, for example.

We thank the reviewer for this careful check. This is not a fitting of any mathematical model. Therefore, we have changed to linear segments as the reviewer suggested in the revised manuscript.

The line in supplementary figure 8 does not pass through the origin (0,0) which I find unlikely to be physically sensible because, as it stands, it would mean negative retention time for pulses shorter than ~10 ms!

We thank the reviewer for this careful observation. Pulse width less than 10 ms simply

cannot turn on the device (Fig. 2d, reproduced as Fig. R6a below). This means that for $t \leq 10$ ms, the retention is essentially *zero* and flat. In the linear fitting, we did not include this flat region. In fact, the fitting yielded an extrapolated *zero* retention at ~ 10 ms input pulse width (Fig. R6b), which is expectedly consistent with experimental result.

To avoid confusion, we have added following explanation in Supplementary Figure 8 (now FigS9): “...Note that this linear fitting does not pass the origin, because only pulses with width >10 ms can turn on the device (Fig. 2d). The extrapolated 0 ms retention at ~ 10 ms input pulse width is consistent to the experimental result (Fig. 2d)”

Finally, there appears to have been little care in preparation of text with multiple typos or imprecisions: for example, p1, line 71 *G sulfureducens* is a bacterium. Referring to it as 'organism' is unnecessarily imprecise.

We have replaced the word 'microorganism' with 'bacterium'. We now have thoroughly checked the languages in the manuscript to ensure a precise presentation (the reviewer can tell our efforts in a Tracked version uploaded together). We thank the reviewer for the time and helpful suggestions. We also wish the reviewer a happy holiday.

Reviewer #2 (Remarks to the Author):

In this work the authors show that a memristor based on protein nanowires can operate with low voltages near the biological potential of 40-100 mV. This potentially reduces power and can be implemented in direct communication with biology.

The paper is well written and reads well. Figures are clear and the story is to the point. However, to warrant publication I do have some comments to be addressed.

We greatly thank the reviewer for confirming the value/potential of the research and the quality of the scientific presentation. We also appreciate the reviewer's suggestions/comments to help strengthen the work. Below please find our detailed responses to the questions.

My main concern is the inherent decrease in stability by decreasing the switching voltage. To my knowledge a low switching (writing) voltage will also mean that by probing (reading) the device, the conductance state can be (slightly) modified. There is a tradeoff between read and write energy and stability of the state, specifically when introducing multiple conductance states. This is also apparent when considering the mechanism of the leaky-integrate and fire neuron demonstrated in Fig 4b. Can the authors comment on this?

We appreciate the reviewer's general view over memristor stability. Please allow us to start with some general perspectives before getting to the specific study here.

Mechanistically, memristors can be classified into electrochemical metallization cells (EMC) and valence change memory (VCM) devices. The memristors that fit the mathematical model were generally VCM devices, in which the analogous conductance change can be described by the continuous drift of oxygen vacancies (*Nature* 2008, 453, 80). In such model, any signal input in principle can cause the drift in vacancy thus the conductance change. In a real device, a low activation energy is still associated with the oxygen vacancy (*Sci. Rep.* 2019, 9, 17019), so a sufficiently low reading voltage (*e.g.*, smaller than the activation energy) is expected to cause negligible state change. But it is a legitimate concern that a reduced writing voltage generally means a reduced activation energy, so the reading voltage has the increasing chance to perturb the written state.

The EMC memristor, on the other hand, is more of a 'threshold' device featuring abrupt conductance change that corresponds to the physical rupture or re-bridging of a metal filament. This also means that static multiple conductance states are less feasible in EMC devices. A reading voltage below the 'threshold' voltage can hardly change the conductance. This is because even though the filament formation is based on field-driven ion migration, the cation ions are not readily available (*v.s.* the readily available oxygen vacancy in VCM). Instead it requires electrochemical redox process (*i.e.*, metal oxidization/reduction), which is largely a threshold event by overcoming the electrochemical reaction potential (*Appl. Catal. B* 2017, 202, 217). This has been experimentally shown in other study that even though the switching voltage in EMC was low (*e.g.*, ~0.2 V), the continuous reading did not alter the conductance state (*Nano Lett.* 2019, 19, 2411).

Our device falls into the EMC category, more precisely, is a volatile EMC memristor. The continuous conductance change in the device is a dynamic modulation that only happens with writing voltage pulses (*e.g.*, Fig. 4d, Fig. 5b), because the dynamic filament evolution still requires a threshold 'reduction' step ($\text{Ag}^+ \rightarrow \text{Ag}$) as illustrated in Fig. 4b. A reading voltage smaller than the writing voltage cannot overcome the reduction overpotential, and hence is not expected to alter the dynamic conductance state.

Another comment is regarding the claim that these low voltage devices open up a path for directly communicative electro-bio interfaces. If this is one of the main applications (apart from lower power consumption), can the authors further comment on the toxicity of silver in bio applications as well as stability of these devices in a biological environment?

We thank the reviewer for a visionary thought over the future potential. This concern can be viewed from two perspectives. First, Ag is applied in practical medicine such as eye treatment and treatment of skin ulcers (*Int. J. Res. Pharm. Sci.* 2017, 4, 1). It is also used as antibacterial material for wound treatment and surgical instruments (*J. Biomed. Mater. Res. A* 2012, 100, 1033). Of course, excessive Ag absorption can cause organ disruption and potential diseases. The memristor contains minimal trace of Ag (e.g., $\sim\mu\text{g}$ in 1 million devices) that is far from biosafety concern. In fact, it is more of a realistic concern from device perspective that, if Ag were dissolved, whether the memristor would still maintain the function.

Both concerns can be addressed by sealing the device body while only exposing the terminals to interface biological tissue for signal retrieval and feedback. In this case, the interconnects or terminals no longer need to be made of Ag, but can be of inert metals such as Pt/Au shown to be biocompatible. There are mature technologies to integrate electronics in sealed flexible substrates for tissue interface (*Nano Lett.* 2017, 17, 5836). For example, there is study showing that Ag memristors were integrated in a skin interface and maintained the functionality using such strategy (*Nat. Nanotechnol.* 2014, 9, 397).

In the revised manuscript (page 8), we now include a brief discussion of the potential strategy for electro-bio interface: "...For example, bio-voltage memristors and neuromorphic components may be integrated in flexible substrates^{53,54} for tissue interfaces, enabling on-site signal processing for close-loop bioelectronics."

The second part of the paper introduces a neuron functionality similar to a Leaky-Integrate-and-Fire. This is an interesting concept and perhaps I am misreading this part, but if the neuron device is not activated anymore, will the device go back to its original state as a real neuron would do? But does that not imply the new conductance state has a low stability? And how does that relate to the memristor state stability.

We think that there is little misunderstanding here. The memristor described here is a volatile one, as it naturally decays to Off if the writing input is removed (Fig. 2d). Such type of memristors are also referred to as 'diffusive' memristors and shown to be important in encoding the relative temporal information in neuromorphic systems (*Nat. Mater.* 2017, 16, 101). In this regard, the memristor naturally goes back to a low-conductance or 'rest' state after the firing, just like the repolarization process in a real neuron.

Fig. R1. (a) Circuit diagram of an artificial neuron. (b) Integrate-and-fire in the artificial neuron (blue spike), after which the neuron naturally decays to a 'rest' state.

A closer version is to add a series resistor (R) and a parallel capacitor (C) to facilitate the

‘repolarization’ process (Fig. R1a). When the memristor is turned on, it quickly discharges the capacitor to bring down the terminal voltage (V_m). At the same time, the series resistor R and the low-resistance memristor (R_{ON}) form a voltage divider, so input pulse is attenuated across the memristor (e.g., by a factor of $R_{ON}/(R_{ON}+R)$) and below the writing threshold. This forms a period like the ‘refractory’ period in a biological neuron after firing. Therefore, the memristor naturally decays to Off after the ‘firing’ (Fig. R1b), before it begins to integrate the next round of pulses—just like a biological neuron.

In the revised manuscript, we now **have added this new version of artificial neuron (Fig. R1) to a new Fig. 6 with corresponding (above) descriptions (Page 7-8).**

Furthermore, if the protein nanowires facilitate the cathodic reduction of Ag^+ , does that mean that only tuning the conductance one way is improved? Or how does this translate to conductance tuning the other way? From the graph (Fig 2b) it looks like both up and down tuning initiates at low voltages.

Theoretically, the electrochemical process in a writing/set process ($Ag \rightarrow Ag^+ \rightarrow Ag$) is the same as the one in an erasing/reset process ($Ag \rightarrow Ag^+ \rightarrow Ag$), with the former moving Ag from electrode to filament (turn on) and the latter moving Ag from filament to electrode (turn off). If the cathodic reduction is the determining step and nanowire facilitates the step, we would still expect a same facilitation in the reset process due to the symmetry (note that the cathode and anode also swap in the two processes).

In reality, as mentioned before, the memristor described here is a volatile one. The turn-off process is not driven by electrical input, but by the interfacial energy relaxation in the filament (*Nat. Mater.* 2017, 16, 101), i.e., through a natural decay (Fig. 2d). Fig. 2b essentially describes (only) the turn-on processes in both polarities (due to a symmetrical planar device structure). As a result, the cathodic facilitation from protein nanowire in principle would work in both processes, whereas here only needs to work in the turn-on process.

We thank the reviewer for the time and helpful suggestions. We also wish the reviewer a happy holiday.

Reviewer #3 (Remarks to the Author):

The authors present an experimental demonstration of a novel memristor device based on ionic silver switches whose formation is mediated by protein nanowires. These devices are shown to have a low operation voltage that is around 100 mV. These voltages are similar to the voltages that occur in biological systems. The demonstration of memristive devices operating as such low-voltages is impressive and is the notable result in this report.

However, this demonstration alone is not a significant enough advancement to merit publication in a high-profile journal such as Nature Communications.

We thank the reviewer for confirming the device novelty and the impressive result in sub-100 mV function. We believe that the research has also provided a concept change in the field and vocalized (i) that neuromorphic emulation can be beyond the functional level and toward bio-parameter matching, and (ii) a catalytic concept for broad engineering strategy. These ideas can lead to new frontiers in pursuing bio-voltage electronics or neuromorphic-bio interfaces. And we believe that such voice and impact live up to *Nature Communication* standard (as was also inferred from the other reviewers).

We do take the reviewer's points seriously and have further demonstrated the unique potential in applications. The revised version now shows both novelty in device and impact in potential applications, providing a more strengthened and comprehensive study for *Nature Communication* standard. Below please find our detailed responses to the questions.

The authors do not provide sufficiently compelling experimental support for their proposed mechanism. The authors propose a three-step mechanism dominated by cathodic reduction for the switching observed in these devices, and they show that protein nanowires can be key to lowering the switching voltage. However, they do not unambiguously demonstrate that a change in the electrochemical behavior of the system is lowering the switching. Figure 3d shows cyclic voltammetry for the Ag on the wire system, but there is effectively no electro chemistry occurring on the SiO₂ only samples; thus, the "shift" is not convincing.

We believe it is because of the less clear plot in Fig 3d (e.g., diminished peaks due to scaling) that gave the impression that 'there is effectively no electrochemistry occurring on SiO₂ only samples'. We have now replotted the curves in different scales (Fig. R1), which we believe can now clearly show the 'shift'.

We provide some explanation to the evidence. In electrochemistry, it is standard to coat different materials on electrode to study the catalytic effect. Both the shifts in the redox peak (*Chem* 2019, 5, 2429) and starting position of current increase (*Adv. Mater.* 2018, 30, 1707319) are used to show facilitation. The new figure shows no shift/facilitation in oxidation peak (gray dashed line), but a shift/facilitation in reduction. The less apparent reduction peak in SiO₂ was consistent to other SiO₂-coated cyclic voltammetry curves (*Coating* 2019, 9, 487; *Nano Biomed. Eng.* 2018, 10, 156). So here we used the shift in the starting position of current increase as the indication (black dashed line).

In the revised manuscript, we now have replotted Fig. 3d (as shown in Fig. R1 here) to show a clear shift. We thank the reviewer's help in improving the clearness in presentation.

Fig. R1. Cyclic voltammetry using nanowire-coated and SiO₂-coated electrodes. The arrow shows the shift in the starting position of current increase.

In addition, is it necessary to use protein nanowires? or is it simply an effect of having transport matrix present? Would a CNT matrix or network of other nanowires work just as well?

We thank the reviewer for bringing about other possible controls to further confirm the protein-nanowire's unique role.

We now have performed controls using both single-walled CNT matrix and Si-nanowire network. Specifically, devices using CNT-matrix initially showed high-conductance transport dominated by CNT (Fig. R2a), whereas devices using semiconducting Si-nanowire network showed low-conduction like bare SiO₂ dielectric (Fig. R2b). None of them formed low-voltage switching following the same forming process. The results are consistent with previous studies (*App. Phys. Lett.* **2017**, 111, 153504; *J. Appl. Phys.* **2018**, 124, 152118), in which Ag memristors using nanowire mesh/network did not yield low-voltage switching (e.g., >1 V). These 'negative' controls indicate that it is not the percolation material structure that contributes to the effect, which is a further indirect support to the catalytic effect from the protein nanowires.

Fig. R2. IVs from devices using (a) CNT and (b) Si-nanowire network.

In the revised manuscript, we have now added these results to a new Supplementary Fig. 17 as negative controls to support protein-nanowire's unique role.

A tremendous amount of Ag is involved in the switching, does changing the surface potential of the SiO₂ with a surfactant change the switching voltage?

The initial forming may have consumed good amount of Ag, which nonetheless is typically observed in other Ag-based memristors (*Adv. Funct. Mater.* **2014**, 24, 5679; *J. Appl. Phys.* **1976**, 47, 2767). We thank the reviewer for the careful thought that the Ag filament may contact SiO₂ surface and hence the surface property of SiO₂ might have contributed to the bio-voltage switching.

We therefore performed controls in which the SiO₂ surface was functionalized with different surface potential. We used both anionic (α -NH₂, ω -COOH-terminated polyethylene glycol) and cationic (ω -Amino-terminated poly(ethylene glycol) methyl ether) surfactants to change the surface charge states and hence surface potential in SiO₂ before depositing protein nanowires. Both devices showed the same bio-voltage switching as protein-nanowire devices made on pristine SiO₂ (Fig. R3). SiO₂ surface functionalization alone (without protein nanowire) could not yield bio-voltage memristive switching.

Fig. R3. Bio-voltage switching in devices with SiO₂ functionalized with (a) anionic and (b) cationic surfactants.

We believe that this is another set of controls to show the contributing role from the protein nanowires. In the revised manuscript, we now added Fig. R3 to a new Supplementary Fig. 16 as further support to nanowire's role.

More critical in my mind for showing a significant advancement would be a demonstration of an end application that requires the low switching voltage. For example, there is no demonstration of a direct connection to a biological system or an ultra-low powered neuromorphic circuit implementing a useful search or visualization operation. There is not even a demonstration that these memristors can be scaled to large systems. Without a demonstration of an impactful application that is enabled by these low-voltage devices, the paper does not make the impact necessary to warrant publication in *Nature Communications*.

We thank the reviewer for the critical thought. While the work focuses on device, we now have also demonstrated the potential in implementing the bio-voltage memristors as described as follows (page 7-8):

“Finally, we show the potential of implementing the bio-voltage memristors in biointerfaces. Various electronic devices such as intracellular bioprobes⁵⁰ and self-powered wearable sensors^{51,52} have been developed to record physiological signals. The recorded signals are generally small and often in the sub-100 mV range,⁵⁰⁻⁵² which require amplification before conventional signal processing. This pre-processing adds to the power and circuitry requirements for future closed-loop bioelectronic interfaces or biomimetic systems. The bio-voltage memristor provides the possibility for direct bio-signal processing to reduce the power and circuitry budget, which is highly desirable for improved sustainability and reduced invasiveness in bio-integrated systems.

Fig. 6a shows the circuit of an artificial neuron with tunable integrate-and-fire response.¹⁴ The input pulses gradually increase the voltage across the memristor (V_m) by charging the capacitor (C) through the resistor (R). The equilibrium voltage peak is dependent on the input frequency relative to the time constant (RC). If a threshold voltage is reached, the memristor will be turned on and transits to low resistance (R_{ON}). If R_{ON} is considerably smaller than R, it will discharge the capacitor to lower V_m . Meanwhile, the input pulses across the memristor will also be attenuated (e.g., by a factor of $R_{ON}/(R_{ON}+R)$ through the R- R_{ON} voltage divider) and below the threshold. This forms a ‘refractory’ period similar to that in a biological neuron after firing. Therefore, the memristor naturally decays to Off after the firing, before it starts to integrate the next round of pulses. The frequency-dependent firing in the artificial neuron can enable artificial bio-reporter to monitor bio-signal changes.

In a proof-of-concept demonstration, emulated biosensing signals⁵⁰⁻⁵² (e.g., 80 mV pulses) were input to the artificial neuron (Fig. 6a). The RC constant (~1 s) was tuned to be close to the period in

normal heart rate (e.g., $f=1.16$ Hz), so the competing charging and discharging yielded a peak V_m below the threshold and could not trigger the neuron firing (Fig. 6b). Abnormal heart rate (e.g., $f=3$ Hz) increased the charging rate to yield a V_m larger than threshold and triggered the neuron firing (Fig. 6c). The firing showed a stochastic feature (Fig. 6d). The artificial neuron here hence realized an on-site health reporter, showing the potential of using bio-voltage memristors to do direct bio-signal processing. Cellular signal (e.g., action potential) has similar amplitude,⁵⁰ and thus the bio-voltage memristor may also enable artificial interneuron for direct cellular communication.”

Please note, that I find no major technical flaws in the material that is presented, and I suggest that the paper is submitted for publication to a more technically focused journal.

We thank the reviewer for confirming the solid research. We believe that the device novelty, now combined with above demonstrated potential in applications, shows the broad impact suitable for *Nature Communications*.

Following are some minor features of the paper that when improved will make the paper easier to understand and will help it have a bigger impact on the community.

There are many small grammatical errors throughout the paper. The readability of the paper would be greatly improved if a thorough edit was performed with an eye towards making it grammatically correct. By no means am I going to illustrate all the errors, but I'll show a couple of examples here:

1st paragraph: “... there has been emerging attempts to use memristors ...” (“has” should be “have” and attempts don't emerge: “... there have been attempts to use memristors ...”

Page 6, 2nd paragraph. This is an important paragraph in the paper, and very challenging to understand as written due to a combination of poor grammar, challenging concepts, and poor ordering of the introduction of concepts. In particular.

“As neural firing is triggered by membrane potential (V_m) directly related to net cytosolic charge ($Q = C_m \cdot V_m$; C_m is the membrane capacitance), the state dynamics is often modeled by $C_m \frac{dV_m}{dt} = I - g_m V_m$ (Eq. 1), where I denotes injection current and $g_m V_m$ the leaky current related to the membrane conductance g_m (Fig. 4a).”

Q is not defined. If I understood this complex sentence properly, one way that could be written is as follows.

Neural firing is triggered by the membrane potential (V_m) which is directly related to the net charges (Q) in a given cytosolic volume ($Q = C_m \cdot V_m$; C_m is the membrane capacitance); therefore the state dynamics are often modeled⁴³ by $C_m \frac{dV_m}{dt} = I - g_m V_m$ (Eq. 1), where I denotes injection current and $g_m V_m$ the leaky current related to the membrane conductance g_m (Fig. 4a).

We thank the reviewer for the frank criticism to the grammatical presentation. It was our negligence not to do the most careful check before submission. We have now not only corrected the grammatical errors the reviewer pointed out, but also carefully gone through the manuscript to correct others (the efforts can be seen by looking at a Tracked version uploaded together).

We are particularly thankful to the reviewer for the helps in rewording the description of the neuron model, which is precise and easier to understand. We thus have used the reviewer's suggested version in the revised manuscript.

As this paper is primarily concerning electronic device behavior, it would probably be helpful to explain cytosolic. The paragraph below the one containing the above sentence does

explain the hypothesis “that the dynamics of filament formation in the memristor (Fig. 4b) is qualitatively analogous to the depolarization in a biological neuron”. But the explanation comes after the initial paragraph that creates the confusion.

We now have revised the manuscript to start with the description of biological process and then mechanistic analogy in memristor, before the statement of constructing artificial neuron (page 6).

Because both cyclic voltammetry (CV) and capacitance (C) are referred to in this paper, it is important that the authors always make it clear which they are referring to. I suggest that they write out the words cyclic voltammetry in the figure caption to Fig. 3.

Great point. We now have skipped the use of acronym for cyclic voltammetry to avoid the confusion.

I don't see where Figures 4A and 4B are referred to in the body of the text. What is their purpose?

We now have changed the capital “A” “B” to “a”, “b” in Fig. 4. Figures 4a and 4b were initially referred in (lines 171 and 178) and (line 177) in the main text. They were placed together to schematically show the similarity in the integrate-and-fire process between a neuron and a memristor.

Be sure that the supplemental figures have complete figure captions or accompanying notes that ensure that the presented data and its technical importance can be fully understood. Following are a few issues that were easily noticed.

SFig 4. & 5 & 7 & 8. &17. Be sure to indicate what device structure (planar or vertical) is used for these data.

4 - protein-nanowire device (nothing indicated geometry)

5 - protein-nanowire memristor (nothing indicated geometry)

6 - protein-nanowire memristors (picture indicates geometry)

7 - protein-nanowire memristor (nothing indicates geometry)

8 - protein-nanowire device (nothing indicates geometry)

9 - vertical protein-nanowire memristors (name and picture indicate geometry)

17 - protein-nanowire memristor (also protein-nanowire synapse) (nothing indicates geometry)

We now have added all the missing device details to all SI figures.

Supplementary Fig 10. Right. The vertical devices are devices are effectively two devices in parallel. What is the appropriate scaling factor? (It doesn't matter here because they are all treated similarly, but what is the appropriate length for comparing vertical to horizontal devices?)

As the reviewer pointed out, a vertical device is effectively two planar devices in parallel. Planar devices had the same width in electrode $\sim 1 \mu\text{m}$ (Fig. 3a), and the vertical devices had equivalent widths ($\times 2$) from $4 \mu\text{m}$ to $40 \mu\text{m}$ (Supplementary Fig. 10 (now SFig. 11)).

There is one difference. Vertical devices had an electrode spacing $\sim 20 \text{ nm}$ defined by the SiO_2 layer thickness, whereas planar devices had a range from 100 to 500 nm due to lithographic limit (Supplementary Fig. 7). For filamentary memristive switching, it was expected that the

switching voltage would be independent of the electrode spacing (the forming voltage would be dependent), which was consistent with the experimental observations (Supplementary Fig. 11b & Fig. 14a).

To push the limit, we have made a vertical device with an electrode size of $\sim 100 \times 100 \text{ nm}^2$ (Fig. R5a), which is equivalent to a planar device with 200 nm electrode width and 20 nm electrode spacing. So it is equivalent to a 5-fold down scaling from the typical planar devices of 1 μm width and 100 nm spacing. The device maintained the sub-100 mV switching after forming (Fig. R5b).

We have now added Fig. R5 to a new Supplementary Fig. 12 to show the further support to the filamentary mechanism and the potential in device scaling.

Fig. R5. (a) SEM image of a vertical $100 \times 100 \text{ nm}^2$ device. (b) Switching IVs (red is the forming curve).

What is the difference in the conditions for SFig. 11 and SFig13.?

A chemical ethanolamine was used during the purification of nanowires, although it was subsequently removed through dialysis. We intended to strictly exclude the possibility of effect coming from ethanolamine residue, by treating the device with normal concentration of ethanolamine (SFig. 11, now SFig. 13). SFig. 13 (now SFig. 15) was a bare device without any treatment.

SFig. 16 (b) What does the data represent? Is it the initial formation switch for five separate devices? Is it the first five cycles of a single device?

SFig. 16 (b) (now SFig. 20) is the first five cycles from a single device. The device never went down to a set voltage below 0.5 V. We have now revised the caption to avoid the confusion.

In summary, we hope our extensive efforts have substantially improved the manuscript to *Nature Communication* standard. We are truly grateful to the reviewer for the constructive comments, in which we can tell the goodwill in helping to improve the work. We also wish the reviewer a great holiday and happy new year.

Reviewers' comments:

Reviewer #1 (Remarks to the Author):

The authors have done an excellent job to address my previously comments. I have no more concerns for this work.

Reviewer #2 (Remarks to the Author):

The authors have responded with care to many of my concerns. There are however a few remaining issues, the first two being quite important:

- 1) The authors have still not mapped the phenotypes observed in the mutant strains shown on figure 5d to the genes of interest. They only verified the nature of the genetic lesions. They could perform rescue experiments, backcross the mutation to a standard genetic background, or at least perform complementation experiments with either other mutants or deficiencies.
- 2) The statement on p.10 about validation of rhythmic phosphorylation is rather vague. It seems the authors attempted to test multiple phosphorylation sites, but show results for only one. It is great that indeed SGG phosphorylation is rhythmic, but what happened with the other attempts at validation? How many sites were tested, and how many were rhythmic, not rhythmic, undetectable? This is important to evaluate the quality of the kinome data.
- 3) The authors wrote on p.13 "Given that JNK1 and JNK2 phosphorylate BMAL1, the mammalian ortholog of *Drosophila* CYC, we hypothesize that BSK can also act on CYC". However, in their response, they indicated that the phosphorylation sites for JNK1/2 on Bmal1 are not conserved in CYC. This weakens their reasoning.
- 4) Figure legend in 3I should refer to data on panel H, not G

Reviewer #3 (Remarks to the Author):

The authors have addressed my concerns and I am happy to recommend this manuscript for publication.

Detailed Responses to Reviewers' Comments

Reviewer #2:

1. The authors have still not mapped the phenotypes observed in the mutant strains shown on figure 5d to the genes of interest. They only verified the nature of the genetic lesions. They could perform rescue experiments, backcross the mutation to a standard genetic background, or at least perform complementation experiments with either other mutants or deficiencies.

We have now backcrossed hep^{G0107}, gish^{EY06451} and Asator^{KG05051} mutant alleles onto an isogenic background. Only Asator mutants retained their circadian phenotype, which means the phenotypes observed in hep and gish mutants are likely due to differences in genetic background of these mutant lines vs. the control. The precise lesion site for nonC¹ mutant has not been characterized, therefore we were not able to backcross it. Instead, we tested another nonC mutant, nonC^{G1076}, which has a P element inserted into the coding sequence of exon 2 and is likely to be a null allele. However, this mutant does not exhibit significantly altered rhythm, indicating that the reduced power observed in nonC¹ mutants is caused by contributing factors in the genetic background of this line rather than disruption of the nonC gene. The behavioral data for hep, gish and nonC mutants are now in Table S7, including previous results acquired on non-isogenic background and new results on isogenic background. We have added description regarding this in the Results and Discussion section on Pg.12, 16 and 17 (marked in red).

2. The statement on p.10 about validation of rhythmic phosphorylation is rather vague. It seems the authors attempted to test multiple phosphorylation sites, but show results for only one. It is great that indeed SGG phosphorylation is rhythmic, but what happened with the other attempts at validation? How many sites were tested, and how many were rhythmic, not rhythmic,

undetectable? This is important to evaluate the quality of the kinome data.

For our last revision, we focused on the relatively well-studied phosphorylation sites (p-sites) with antibodies available. Besides SGG-Y214, we also examined BSK-Y185. However, we did not observe significant oscillation of phosphorylation likely because the antibody we used detects phosphorylation at both T183 and Y185. There is no antibody that only binds phosphorylated Y185. To further validate our cyclic phosphoproteome data, we have now systematically searched for antibodies for all 789 oscillating p-sites as well as antibodies for their corresponding proteins (for normalization purposes). We have found and acquired antibodies for p-HSP83-S219 and HSP83 protein (i.e. antibodies against mouse p-HSP90-S223 and HSP90). These antibodies have been used in mouse 3T3 cells but unfortunately we were not able to detect any signal with the p-HSP83-S219 antibody. Please see below for our result.

3. The authors wrote on p.13 "Given that JNK1 and JNK2 phosphorylate BMAL1, the mammalian ortholog of *Drosophila* CYC, we hypothesize that BSK can also act on CYC". However, in their response, they indicated that the phosphorylation sites for JNK1/2 on Bmal1 are not conserved in CYC. This weakens their reasoning.

We thank the reviewer for pointing this out. We have now removed this sentence from the manuscript as well as the hypothetical arrow from BSK to CYC in Fig. 6a.

4. Figure legend in 3I should refer to data on panel H, not G.

Thank you for pointing this out. We have fixed it.

Reviewers' comments:

Reviewer #2 (Remarks to the Author):

The authors have addressed my comments with additional text corrections and experiments with backcrossed mutations. These experiments now indicate that for two of the three classical/insertional mutants with a phenotype, the circadian phenotype was not caused by the mutation of interest. This is not very surprising as the phenotypes were quite mild. The Asator mutant is the last to be still presented in the main figure as a positive hit. The phenotype is weaker than initially reported now that the mutation was backcrossed. This was however done for only two generations, while 6-generation backcrossing is standard. I am therefore not convinced that the 0.6-hr long period phenotype and lower amplitude of rhythms would still be visible after having more thoroughly backcrossed the Asator mutation.

This said, I think this manuscript should be published as it contains really important information for the field of chronobiology. Besides the multi-omic studies, which are very original, the kinase screen provides interesting new circadian kinases. The screen just needs to be presented more clearly to the reader, with three groups of candidate circadian kinases. Group 1 would be genes with solid evidence (multiple RNAi lines) and include GSKT, Dsor1 and CK1alpha. Group 2 would be genes that perhaps are involved in the clock (BSK, GISH, HEP), for which only overexpression gives a phenotype. Group 3 would be genes for which there is currently no clear evidence, with the assays that were used here (they could be important for other circadian functions than locomotor rhythms). Group 3 should include Asator. 6 RNAi lines were tested for this gene as well as 4 mutants. Only one mutant shows a weak phenotype, which may not survive further scrutiny. I would suggest moving results with this mutant to the supplemental data and acknowledging that the evidence for Asator is rather weak at this point. I would also be quite brief with the classic/insertional mutants since at the end none of them gave clear phenotypes caused by the gene of interest. It can be briefly mentioned that three of them gave initially a phenotype, but that after two generations of backcrossing, these phenotypes disappeared or were weakened. I would also change the number of circadian kinases from 14 to 10 to reflect those for which evidence is solid, based on past and the current studies

We thank the reviewer for his/her thoughtful comments and suggestions. We have addressed these comments and suggestions as described below. The original reviews are listed and our responses are in italic font. Edits made in the text of the manuscript are marked in red.

Referee #2:

The authors have addressed my comments with additional text corrections and experiments with backcrossed mutations. These experiments now indicate that for two of the three classical/insertional mutants with a phenotype, the circadian phenotype was not caused by the mutation of interest. This is not very surprising as the phenotypes were quite mild. The Asator mutant is the last to be still presented in the main figure as a positive hit. The phenotype is weaker than initially reported now that the mutation was backcrossed. This was however done for only two generation, while 6-generation backcrossing is standard. I am therefore not convinced that the 0.6-hr long period phenotype and lower amplitude of rhythms would still be visible after having more thoroughly backcrossed the Asator mutation.

This said, I think this manuscript should be published as it contains really important information for the field of chronobiology. Besides the multi-omic studies, which are very original, the kinase screen provides interesting new circadian kinases. The screen just needs to be presented more clearly to the reader, with three groups of candidate circadian kinases. Group 1 would be genes with solid evidence (multiple RNAi lines) and include GSKT, Dsor1 and CKIalpha. Group 2 would be genes that perhaps are involved in the clock (BSK, GISH, HEP), for which only overexpression gives a phenotype. Group3 would be genes for which there is currently no clear evidence, with the assays that were used here (they could be important for other circadian functions than locomotor rhythms). Group 3 should include Asator. 6 RNAi lines were tested for this gene as well as 4 mutants. Only one mutant shows a weak phenotype, which may not survive further scrutiny. I would suggest moving results with this mutant to the supplemental data and

acknowledging that the evidence for Asator is rather weak at this point. I would also be quite brief with the classic/insertional mutants since at the end none of them gave clear phenotypes caused by the gene of interest. It can be briefly mentioned that three of them gave initially a phenotype, but that after two generations of backcrossing, these phenotypes disappeared or were weakened. I would also change the number of circadian kinases from 14 to 10 to reflect those for which evidence is solid, based on past and the current studies

As the reviewer suggested, we have now added descriptions regarding the three groups of kinases to the second paragraph of Pg.12 which is marked in red: "Based on the strength of circadian phenotypes observed, we classified the remaining 20 predicted circadian kinases into three groups. Group 1 includes GSKT, DSOR1 and CKIalpha, the phenotypes of which were confirmed by multiple independent RNAi lines and thus are highly likely to regulate the clock. Group 2 includes GISH, BSK and HEP, which only demonstrated phenotypes when over-expressed and thus are potential regulators of the clock. The rest of the predicted circadian kinases belong to Group 3, as they showed no clear evidence of involvement in locomotor rhythm modulation." We have also made relevant changes in Fig. 4e, 5d, 6 and S10.

We have moved the behavioral results of Asator mutant to Fig.S8c and have simplified the description of behavioral results regarding the mutant strains which is in the first paragraph of Pg.12 (marked in red): "Lastly, we tested 33 mutant or potential mutant lines and found three mutants displayed reduced power or lengthened period (Supplementary information, Fig. S8b, S9 and Table S7). However, after backcrossing these lines onto an isogenic background and/or testing additional alleles, we observed either no phenotype or weaker phenotype, which means the phenotypes previously observed are likely due to genetic background differences rather than defects caused by the mutations (Supplementary information, Fig. S8c and Table S7)." We have deleted discussion regarding Asator as well.

We no longer refer to Group 2 kinases as circadian kinases and have made corresponding edits in the main text, methods, figure and table legends which are all marked in red.